

# Improvement of stratospheric aerosol extinction retrieval from OMPS/LP using a new aerosol model

Zhong Chen[1], Pawan K. Bhartia[2], Robert Loughman[3], Peter Colarco[2] and Matthew DeLand[1]

[1]Science Systems and Applications, Inc., Lanham, Maryland, 20706, USA
[2]NASA Goddard Space Flight Center, Greenbelt, Maryland, 20771, USA
[3]Department of Atmospheric and Planetary Sciences, Hampton University, Hampton, Virginia, 23668, USA

Correspondence to: Zhong Chen (zhong.chen@ssaihq.com)

## Abstract

The Ozone Mapping and Profiler Suite Limb Profiler (OMPS/LP) has been flying on the Suomi NPP satellite since October 2011. It is designed to produce ozone and aerosol vertical profiles at

~2 km vertical resolution over the entire sunlit globe. Aerosol extinction profiles are computed with Mie theory using radiances measured at 675 nm. The operational Version 1.0 (V1.0) aerosol extinction retrieval algorithm assumes a bimodal lognormal aerosol size distribution (ASD) whose parameters were derived by combining an in situ measurement of aerosol microphysics with the SAGE II aerosol extinction climatology. Internal analysis indicates that this bimodal

lognormal ASD does not sufficiently explain the spectral dependence of LP measured radiances. In this paper we describe the derivation of an improved aerosol size distribution, designated Version 1.5 (V1.5), for the LP retrieval algorithm. The new ASD uses a gamma function distribution that is derived from Community Aerosol and Radiation Model for Atmospheres (CARMA) calculated results. A cumulative distribution fit derived from the gamma function

ASD gives better agreement with CARMA results at small particle radii than bimodal or unimodal functions. The new ASD also explains the spectral dependence of LP measured radiances better than the V1.0 ASD. We find that the impact of our choice of ASD on the retrieved extinctions varies strongly with the underlying reflectivity of the scene. Initial comparisons with co-located extinction profiles retrieved at 676 nm from the SAGE III/ISS

instrument show a significant improvement in agreement for the LP V1.5 retrievals. Zonal mean extinction profiles agree to within 10% between 19-29 km, and regression fits of collocated samples show improved correlation and reduced scatter compared to the V1.0 product. This improved agreement will motivate development of more sophisticated ASDs from CARMA results that incorporate latitude, altitude, and seasonal variations in aerosol properties.



## 1. Introduction

Accurate estimation of stratospheric aerosol is important because aerosols in the stratosphere have an important influence on climate variability through their contribution to direct radiative forcing, although the magnitude of this term is still uncertain (Ridley et al.,
2014). Aerosols also play an important role in the chemical and dynamic processes related to ozone destruction in the stratosphere. Therefore, long-term measurement of the distribution of aerosols is necessary for a better understanding of stratospheric processes.

The Ozone Mapping and Profiler Suite Limb Profiler (OMPS/LP) is one of three OMPS instruments onboard the Suomi National Polar-orbiting Partnership (S-NPP) satellite (Flynn et
al., 2007). S-NPP was launched in October 2011, into a sun-synchronous polar orbit. The local time of the ascending node of the S-NPP orbit is 13:30. The LP instrument collects limb scattered radiance data and solar irradiance data on a 2-D charge coupled device (CCD) array over a wide spectral range (290-1000 nm) and a wide vertical range (0-80 km) through three parallel vertical slits. These spectra are primarily used to retrieve vertical profiles of ozone (Rault
and Loughman, 2013), aerosol extinction coefficient (Loughman at al., 2018; Chen et al., 2018), and also cloud-top height (Chen et al., 2016). More details about the OMPS/LP instrument design and capabilities are provided in (Jaross et al., 2014).

Instruments that measure scattered radiation need to assume some form of aerosol size distribution (ASD) to convert their measured information into aerosol extinction. These
instruments include limb scattered instruments such as Scanning Imaging Absorption spectrometer for Atmospheric Cartography (SCIAMACHY) (von Savigny et al., 2015), Optical Spectrograph and Infrared Imaging System (OSIRIS) (Bourassa et al., 2008), OMPS/LP (Loughman at al., 2018; Chen et al., 2018), and space and ground-based lidars, e.g., Cloud-Aerosol Lidar with Orthogonal Polarization (CALIOP) (Winker et al., 2009). By contrast,
instruments that employ solar, lunar, and stellar occultation techniques such as Stratospheric Aerosol and Gas Experiment (SAGE) II (Chu et al., 1989), SAGE III (Thomason et al., 2010) and Global Ozone Monitoring by Occultation of Stars (GOMOS) (Bertaux et al., 2010) can derive extinction directly from their transmission measurements without assuming an ASD.

In this study, we determine a new ASD by calculating a fit to results produced by the
Community Aerosol and Radiation Model for Atmospheres (CARMA) model (Colarco et al., 2003, 2014) in order to improve the accuracy of aerosol extinction profiles retrieved from



OMPS/LP measurements. The revised ASD is used in the new V1.5 OMPS/LP aerosol extinction retrieval algorithm, which demonstrates better performance in internal validation tests (e.g. absolute difference and spectral dependence of calculated radiances vs. measurements) compared to the V1.0 OMPS/LP algorithm. We also validate the revised ASD through

comparisons to independent satellite retrievals of aerosol extinction from SAGE III/ISS solar occultation measurements.

## 2. LP Algorithm Description

The original LP aerosol extinction retrieval algorithm described in (Rault and Loughman,
2013) uses radiance data measured at multiple wavelengths in the visible and near-IR spectral region. The updated version V1.0 algorithm described in detail by (Loughman et al., 2018) is based on Mie theory, using radiances from one wavelength at 675 nm. The aerosol extinction profiles are retrieved from limb-scatter observations using the aerosol scattering index (ASI) as the measurement vector. The ASI is the fractional difference between a given radiance and the
calculated radiance assuming a pure Rayleigh atmosphere bounded by a Lambertian surface. This quantity is roughly proportional to aerosol extinction, as described in (Loughman et al., 2018), and is defined at wavelength $\lambda$ at altitude $z$ in equations (1)-(2):

$$\text{ASI}_m(\lambda, z) = [I_m(\lambda, z) - I_0(\lambda, z)]/I_0(\lambda, z) \qquad (1)$$

$$\text{ASI}_c(\lambda, z) = [I_c(\lambda, z) - I_0(\lambda, z)]/I_0(\lambda, z) \qquad (2)$$

The measured radiance is denoted by $I_m$, while $I_c$ and $I_0$ represent radiances calculated under differing conditions: For $I_c$, the model atmosphere includes the most recently updated aerosol extinction profile, while the model atmosphere is aerosol-free for the calculation of $I_0$. The radiances are normalized (i.e., divided by their value at the normalization altitude, 40.5 km) in all cases, and form the basis for the measured and calculated ASI ($\text{ASI}_m$ and $\text{ASI}_c$, respectively).
Normalizing the radiances reduces the effects of surface/cloud reflectance and errors in sensor absolute calibration. This formulation removes the first order effects of both Rayleigh scattering and reflectivity, though there are second order effects which will be discussed later. As discussed in (Loughman et al., 2018), the effect of Rayleigh and aerosol scattering on radiances is not strictly additive when the optical path along the line of sight (LOS) becomes thick. For example,
Rayleigh scattering also attenuates aerosol scattering, reducing the sensitivity of the ASI to aerosol loading at lower altitudes (i.e., where Rayleigh scattering is high). The observed limb



radiances are strongly affected by diffuse upwelling radiation and Rayleigh scattering along the LOS. ASI is far less affected by these effects, so aerosol signals are much easier to see in ASI.

Assuming that optically thin conditions prevail, the radiance sensitivity is approximately proportional to the change in aerosol extinction ($E$) at the tangent point for the LOS. The V1.5

aerosol extinction retrieval therefore employs a non-linear iterative technique, based on Chahine's non-linear relaxation technique (e.g. Chahine, 1970):

$$x_i^{(k+1)} = x_i^{(k)} \frac{y_i^m}{(y_i^c)^{(k)}} \quad (3)$$

where $x_i^{(k)}$ represents the state vector (i.e., extinction) at altitude $z_i$ after $k$ iterations of the retrieval algorithm. The measurement vector $y_i^m$ represents the measured ASI$_m$ at tangent height

$h_i=z_i$. The Gauss-Seidel limb scattering (GSLS) radiative transfer model (Herman et al., 1994; Herman et al., 1995; Loughman et al., 2004; Loughman et al., 2015) calculates the ASI$_c$ vector $y_i^c$ at each iteration, using the extinction profile given by $x_i^{(k)}$. The cross-section and aerosol scattering phase function are calculated from an assumed ASD using Mie theory assuming spherical droplets of sulfuric acid ($H_2SO_4$). An initial guess for aerosol profile $x_i^{(0)}$ is constructed

by using an aerosol extinction climatology derived from SAGE II data for the period 2000-2004. The V1.0 algorithm performs three iterations of Eq. (3) to reach the final extinction profile. The LP aerosol product provides extinction profiles from cloud top height to 40 km. We flag the lowest level of the retrieved aerosol profile at the cloud-top altitude, which is determined using the algorithm described in (Chen et al., 2016). Additional discussion of the LP aerosol retrieval

algorithm is presented by (Loughman et al., 2018).

## 3. Aerosol Size Distribution

Retrieval of aerosol extinction profiles from limb scattering measurements requires the specification of an aerosol size distribution (ASD) to represent the microphysical properties of

the aerosol particles. Different functional forms can be selected to represent the ASD. The V1.0 LP aerosol algorithm retrieves extinction profiles by assuming a bimodal lognormal size distribution (BD):

$$n(r) = \sum_{i=1}^{2} \frac{N_i}{r\sqrt{2\pi} \ln \sigma_i} \exp\left( -\frac{1}{2} \left[ \frac{\ln(r/r_i)}{\ln \sigma_i} \right]^2 \right) \tag{4}$$





where $n(r)$ is the size distribution function (cm$^{-3}$μm$^{-1}$), $i$ represents the $i$th mode of the distribution ($i$ = 1 and 2 indicates fine and coarse mode, respectively), $r$ is the particle radius (μm), $r_i$ is the median radius, $\sigma_i$ is the standard deviation, and $N_i$ is the number of the particles corresponding to the mode $i$ (in cm$^{-3}$). The fine and coarse mode size parameters of this

distribution (see Table 1) are primarily based on ER-2 measurements made in August, 1991, at 36°N and 121°W and at 16.5 km (Pueschel et al., 1994). Since the observed coarse mode fraction $f_c$ in the (Pueschel et al., 1994) data was very high following the eruption of Mt. Pinatubo, we adjusted $f_c$ downward to provide an Angstrom Exponent (defined in Eq, (5)) of 2.0.

$$AE = -\frac{\ln[E(\lambda_1)] - \ln[E(\lambda_2)]}{\ln(\lambda_1) - \ln(\lambda_2)} \qquad (5)$$

where $E$ is the aerosol extinction at wavelength $\lambda$. We chose AE = 2.0 as our reference because it represents the mean value of AE at altitude 20 km estimated from SAGE II (version 7.0) aerosol extinction data (Damadeo et al., 2013) at 525 nm and 1020 nm taken during the period 2000-2005, when the stratosphere was relatively clean and roughly similar to the present day

stratosphere (Loughman et al., 2018).

The main motivation for using a bimodal size distribution arose from the desire to make comparisons with the existing in-situ optical particle counters (OPC) dataset, which generally features a bimodal size distribution at the altitudes where the stratospheric aerosol extinction is greatest (Deshler et al., 2003). However, specifying the 5 independent parameters (two mode

radii, two mode widths, and the coarse mode fraction) needed to define this more complex distribution can be challenging. Most OPC measurements have no independent information at radii less than 0.1 μm, so that the aerosol size distribution between 0.01-0.1 μm is poorly defined (Kovilakam and Deshler, 2015). The lack of information in the OPC data gap region results in greater uncertainty in fitting data using the bimodal size distribution function (see Appendix A).

As pointed out in Appendix A, the OPC doesn't count particles smaller than 0.1 μm, but the information on the smaller particles comes from super-saturating the cell so that all particles are lit up and one gets this gross count on the total number (with some probably large uncertainty). Although a seemingly arbitrary collection of bimodal size distributions would fit the resolved OPC data equally well the residual uncertainty results in large uncertainty in the phase function.

Moreover, internal evaluation of V1.0 algorithm performance (described in Sect. 4) and



comparison of retrieved extinction profiles with SAGE III/ISS data (described in Sect. 5) also raised questions about the use of the V1.0 bimodal size distribution.

To select an alternate ASD for use in LP retrievals, we have used results from the Community Aerosol and Radiation Model for Atmospheres (CARMA). CARMA is a sectional aerosol and cloud microphysics model that has been used to study a wide variety of problems in planetary atmospheres (Toon et al., 1979, 1988; Turco et al., 1979; Bardeen et al., 2008; Colarco et al., 2003, 2014; English et al., 2011, 2012; Yu et al., 2015). The CARMA model is coupled here to the NASA Goddard Earth Observing System (GEOS) Earth system model, a three-dimensional atmospheric general circulation model, as described in Colarco et al. (2014), and provides simulated aerosol distributions over a full range of latitude and longitude, altitude, and season. Colarco et al. (2014) describes how CARMA was implemented initially for dust and sea salt. The usage in GEOS for sulfate aerosols is a relatively new capability, with the sulfur chemistry mechanism and aerosol microphysics as in English et al. (2011) and as described and evaluated in Aquila et al. (2018 – paper in preparation). The particle size distribution is represented by 22 size bins covering a wide range of radii from 0.000267 μm to 2.79 μm. For this paper, we use output from simulations in which volcanic eruptions have been turned off, so that the background aerosol distribution reflects anthropogenic and non-volcanic sources.

We have chosen the gamma distribution (e.g., Chylek et al., 1992) to describe the size distribution of aerosols for OMPS/LP retrievals (Chen et al., 2018). This function is described in Eq. (6).

$$n(r) = \frac{N_0 \beta^\alpha r^{\alpha-1}}{\Gamma(\alpha)} \exp(-r\beta) \qquad (6)$$

where $n(r)$ is the size distribution function (cm$^{-3}$μm$^{-1}$), $N_0$ is the total number density of aerosols (cm$^{-3}$), $\alpha$ and $\beta$ (μm$^{-1}$) are the fitting parameters, and $\Gamma$ is Euler's Gamma function. At small radii this function follows a power law, while at large radii it follows an exponential function. In contrast to the BD, which has 5 adjustable parameters, the gamma function has only two parameters to be specified, the shape parameter $\alpha$ and the scale parameter $\beta$. These parameters have a unique relationship to the effective radius:

$$r_{eff} = \frac{\int_0^\infty r^3 n(r)\,dr}{\int_0^\infty r^2 n(r)\,dr} = \frac{(\alpha+2)}{\beta} \qquad (7)$$



In order to fit the gamma distribution (GD) to CARMA results, we calculate the cumulative aerosol size distribution,

$$N(>r) = \int_r^{r_{max}} n(r)dr \qquad (8)$$

where $N(>r)$ represents the concentration of all particles larger than $r$. The integral is performed

over a range of sizes from $r_{min}$ to $r_{max}$. The two parameters of the GD are determined by fitting the cumulative distribution function (CDF) of Eq. (8) using a Levenberg-Marquardt nonlinear least squares regression algorithm. The scattering cross sections and phase functions are then calculated using Mie theory assuming spherical particles of refractive index of $1.448 + 0i$, which is the same as that assumed in the LP V1.0 aerosol algorithm.

We created a subset of CARMA results that is approximately consistent with the (Deshler et al., 2003) long-term measurements by averaging June-July-August model results to create a climatology, then extracting aerosol size distribution values for the approximate location (41°N, 105°W) and altitude (20 km) of those measurements. We then calculated CDF fits to the CARMA results using unimodal normal distribution (UD), bimodal lognormal distribution (BD)

and the gamma distribution (GD) functions (described in Eqs. (4) and (6)). For consistency with the OPC database, CARMA bins between 0.01-0.1 μm were excluded from the fit. Figure 1a shows that while these CDF fits are relatively similar, the GD function does give the best agreement with the excluded CARMA values.

Figure 1b compares the derived differential size distributions from the three fits, which

are plotted as $dN/d\log r$ vs. $r$ in log-log scale (here log is the logarithm to base 10). The BD function used for LP V1.0 processing is also shown for comparison. In contrast to the cumulative distribution functions shown in Fig. 1a, the differential distributions differ significantly for $r \leq 0.1$ μm, as well as $r \approx 0.3$ μm. Note that the V1.0 bimodal ASD has the largest $dN/d\log r$ value at $r \approx 0.1$ μm and the smallest $dN/d\log r$ value at $r \approx 0.3$ μm. As a result,

the corresponding aerosol scattering phase function for the BD fit is closer to a Rayleigh phase function at large scattering angles ($\Theta > 120°$), as shown in Fig. 2. Fractional differences of 40% in this region can lead to up to a factor of 2.5 larger extinction values at 20.5 km and at low effective reflectivities $\rho$ (see Sect. 4), where the derived extinctions are roughly inversely proportional to the P($\Theta$), as discussed by (Loughman et al., 2018).





We have selected the gamma size distribution derived from CARMA results in this work to assess the impact of ASD on stratospheric aerosol extinction profile retrieval from OMPS/LP limb measurements. The two fitted parameters ($\alpha = 1.8$ and $\beta = 20.5$) determined from the GD fit at 20 km produce an AE of 2.0 and a $r_{eff}$ of 0.18 $\mu$m. These values match the average values

determined from SAGE II version 7.0 data (Thomason et al., 2008; Damadeo et al., 2013) during the 2000-2005 period. Hereafter, this resultant ASD derived from the CARMA results will be labeled as V1.5 ASD, while the ASD assumed in LP V1 will be labeled as V1.0 ASD. The current LP retrieval algorithm assumes that the size distribution is height independent, so that one function is used to represent the aerosol size distribution at all heights. We plan to use

CARMA model results in a future version of the algorithm to incorporate variation in ASD and P($\Theta$) with altitude, latitude, season and after a volcanic eruption.

Figure 3a shows the impact on the gamma distribution P($\Theta$) of changing the mode parameters by ±10% relative to the baseline mode, for the range of scattering angles viewed during a single OMPS/LP orbit (Chen et al., 2018). It is apparent that the phase function is quite

sensitive to $\beta$. A ±10% change in $\beta$ can produce a ±10% change in the calculated aerosol phase function at moderate scattering angles ($\Theta = 70°-100°$), whereas a ±10% change in $\alpha$ only yields a ±3% change in phase function at $\Theta > 70°$. The changes in P($\Theta$) (Fig. 3a) lead to corresponding significant changes in retrieved aerosol extinctions, as shown in Fig. 3b. The changes in aerosol extinction are approximately anti-correlated with the phase function variations, although smaller

in magnitude. The wiggly structures of the extinction data are caused by the variation in scene reflectivity, which is discussed further in Sect. 4.

It is important to point out that OMPS/LP measurements cover a wide range of scattering angles with a well-defined latitude dependence. Figure 4 shows the variation of $\Theta$ with latitude for two dates corresponding to solstice conditions. Note that high values of $\Theta$ are always

observed in the Southern Hemisphere, while low values of $\Theta$ are observed in the Northern Hemisphere. The impact of this sampling on measured ASI is discussed in Sect. 4.

**4. Results and discussion**

In Sect. 3, we described the creation of the gamma aerosol size distribution model

derived from CARMA results. We have evaluated the impact of the new ASD derived in Sect. 3 on LP aerosol retrievals by reprocessing all LP data for a one month period that encompasses the



Calbuco volcano eruption. This eruption occurred in Chile (41.3° S, 72.6° W) on April 22, 2015, and had a clear impact on the stratospheric aerosol distribution.

Figure 5 shows scatter diagrams of retrieved aerosol extinctions at 20.5 and 25.5 km as a function of latitude for the V1.5 ASD (blue) and from V1.0 ASD (green), as well as their ratios

($E_{V1.5}/E_{V1.0}$, black) for the entire month of data following the Calbuco eruption (Chen et al., 2018).  Extinction values from the V1.5 retrievals (top row) have a similar latitude dependence to the V1.0 retrieval values (middle row) for both 25.5 km and 20.5 km.  However, the extinction ratio ($E_{V1.5}/E_{V1.0}$) decreases in magnitude from high Southern Hemisphere latitudes to high Northern Hemisphere latitudes.  The inverse of the phase function ratio, i.e.

$[P_{V1.5}(\Theta)/P_{V1.0}(\Theta)]^{-1}$ is also shown for comparison in the bottom row of Fig. 5.  The observed change in extinction from the V1.0 ASD to the V1.5 ASD is typically smaller than the corresponding phase function change at all SH latitudes and at NH latitudes less than ~40°N. This difference is caused in part by the "smearing" effect of multiple scattering, which becomes more pronounced at lower altitudes (note the larger scatter of extinction ratio values in Fig. 5f).

The change in extinction ratio is greater than 1.0 at most latitudes due to the change in phase function presented in Fig. 2 and the mapping between LP scattering angle and measurement latitude illustrated in Fig. 4.

Figures 5e and 5f show significantly more variability in extinction ratio of $E_{V1.5}/E_{V1.0}$ at SH latitudes, which is correlated mainly with the variation of effective reflectivity $\rho$. $\rho$ is derived

from the LP measurements at 675 nm to represent Earth surface reflectance. In the LP retrieval algorithm, $\rho$ is determined by comparing the measured data to model radiances at 40 km using a Lambertian surface (Loughman et al., 2018). This "Lambert-equivalent" reflectivity (LER) value typically differs from the true surface reflectivity due to diffuse upwelling radiation (DUR) contributions from clouds, aerosols, and other features within the scene.

As discussed in (Loughman et al., 2018; Chen et al., 2018), the effect of Rayleigh scattering and aerosol scattering on radiances is not strictly additive. Rayleigh scattering also attenuates aerosol scattering, which reduces the measured radiance. This effect increases at lower altitudes, ultimately making the radiances insensitive to aerosol scattering. This behavior as a function of reflectivity is further illustrated in Fig. 6.  This figure shows the extinction ratio data

from Fig. 5 plotted as a function of reflectivity, and binned into 13 latitude bands by color.  The



dependence of the extinction ratio ($E_{V1.5}/E_{V1.0}$) on ρ can become non-linear at low reflectivity, and the slope of the linear portion of this figure (ρ > 0.2) varies with latitude.

While the altitude normalization used to construct the ASI measurement vector in Eq. (1) reduces the effect of DUR in the LP aerosol extinction profile retrieval considerably, there are second order effects present that make ASI sensitive to ρ at altitudes where there are aerosols. This occurs because DUR is scattered by the aerosols at an average scattering angle close to 90˚, while the direct solar radiation is scattered at a wider range of angles shown in Fig. 4. As noted previously, Rayleigh scattering attenuation of aerosol scattering increases at lower altitudes, so that ASI also becomes insensitive to aerosol scattering. For singly scattered (SS) radiances, assuming that the attenuation of SS radiance along the LOS is small, ASI is proportional to the product of aerosol extinction $E$, and P(Θ). So, in this approximation the spectral dependence of ASI should be determined by the spectral dependence of $E$*P(Θ), which is determined by ASD. Hence, if the assumed ASD is correct, the measured and calculated spectral dependence of ASI should be consistent.

In Fig. 7, ASI residuals (difference between the measured ASI and the calculated ASI) from V1.0 and V1.5 retrievals at 20.5 km are plotted as a function of latitude for wavelengths not used in the LP aerosol retrieval (352 nm, 430 nm, 508 nm, 600 nm, 745 nm, 869 nm) for the V1.5 test processing data set. Residuals at the retrieval wavelength (675 nm) are not shown because they are very close to zero for both cases. The residuals produced by the V1.5 ASD are closer to zero than the V1.0 residuals for all wavelengths, indicating that the gamma function ASD more effectively represents the OMPS/LP measurements.

Figure 8 shows the ratio of ASI(745 nm)/ASI(508 nm) at 20.5 km and 25.5 km as a function of latitude, using LP measurements and the calculated ASI values from the V1.0 and V1.5 ASDs. The agreement between measured and calculated ASI ratio is significantly better for the V1.5 ASD, demonstrating the improved representation of spectral dependence with this function. Similar figures can be constructed for other combinations of LP wavelengths. The results shown in Fig. 7 and 8 demonstrate how internal analysis of LP aerosol retrieval results can help identify the most appropriate ASD to use for these retrievals. We note that the ability to distinguish between ASDs is better in the NH, when LP scattering angles are lower and the relative uncertainty in P(Θ) is reduced. We therefore use comparisons with external measurements to obtain additional validation of our choice for the V1.5 ASD.



## 5. Comparison with SAGE III/ISS

The Stratospheric Aerosol and Gas Experiment on the International Space Station (SAGE III/ISS) developed by NASA Langley Research Center (LaRC) was launched to the International Space Station in February of 2017. SAGE III/ISS provides limb occultation measurements of aerosols and gases in the stratosphere and upper troposphere (Chu et al., 1998). The SAGE series of occultation measurements have been extensively evaluated and compared with other space based instruments and have been found to have relatively high precision and accuracy (Bourassa et al., 2012). A general description of the solar occultation measurement technique is provided by (McCormick et al., 1979). The ISS travels in a Low-Earth orbit at an altitude of 330-435 km at an inclination of 51.6°. With these orbital parameters, solar occultation measurement opportunities cover a large range of latitudes (between 60° S and 60° N). The solar occultation measurement Version 5 Level 2 data (available through NASA's Atmospheric Science Data Center) were collected during the period June-December 2017. SAGE III/ISS scientists have released this initial dataset (which includes retrievals of ozone, aerosols and nitrogen dioxide from solar occultation measurements) in order to solicit feedback from the international atmospheric science community. Figure 9 shows the spatio-temporal coverage of the available datasets used for this study. Profiles of aerosol extinction at 9 wavelengths reported by SAGE III/ISS (384.2, 448.5, 520.5, 601.6, 676.0, 756.0, 869.2, 1021.2, 1544.0 nm) are provided from the surface or cloud top to an altitude of 45 km, with a vertical resolution of 0.5 km at the tangent point location. We have compared OMPS/LP aerosol extinction retrievals at 675 nm and SAGE III/ISS aerosol extinction retrievals at 676 nm directly, using SAGE III samples that correspond to the OMPS/LP 1 km altitude grid. For OMPS/LP, only data from the center slit were taken into consideration, and all data below the cloud height were rejected.

Figure 10 shows time series of OMPS/LP and SAGE III/ISS extinctions for six 10° latitude bins at 20.5 km from June to December 2017. Red and black dots show individual measurements from sunrise (SR) and sunset (SS), blue and green dots represent LP V1.5 and V1.0, pink and yellow lines are the median of the individual LP extinctions, respectively. Note that a lot of the blue dots are lying under green ones. SAGE III/ISS data is available for relatively few days, while OMPS/LP measurements provide daily global coverage. This time series demonstrates that reasonable agreement is observed between LP V1.5 and SAGE III/ISS datasets, while LP V1.0 retrievals typically show lower extinction values except in the Northern



Hemisphere during winter. This difference is consistent with the scattering angle dependence discussed in Sect. 4, since LP measures at small scattering angles at this time. Pattern differences between LP V1.5 and V1.0 reflect the impact of aerosol size distribution on aerosol extinction. Large variability throughout most of the time series can be explained by differences

in spatial or temporal sampling. The enhanced extinctions at 20.5 km observed by both LP and SAGE III in September–December in the Northern Hemisphere are likely associated with the enormous pyrocumulus events caused by the British Columbia wildfires in August 2017.

Further indication of the level of agreement between OMPS/LP and SAGE III/ISS is provided by comparing the zonal average profiles. For this comparison, a relative broad

collocation requirement of ±5° latitude was used, and longitudinal differences were ignored in order to maintain a minimal comparison set size. The variation in SAGE III sampling illustrated in Fig. 9 limits each zonal average to 2-3 consecutive days of measurements. The data have been binned and averaged to the OMPS/LP reporting altitudes for direct comparison.

Figure 11 shows the zonal mean extinction profiles between 15 km to 30 km altitude in

six ±5° latitude bins. For each latitude bin, the mean OMPS/LP profile is usually composed of between 180 and 400 measurements, while the number of SAGE III profiles used in each average is much smaller, typically between 20 and 40 profiles. For all latitude bins, the general agreement between LP V1.5 and SAGE III is quite good except at lower altitudes (< 18 km), where larger differences may indicate limitations in the LP retrieval algorithm. In contrast, LP

V1.0 retrievals are systematically lower than SAGE III over the entire profile. This improvement in agreement gives us confidence that the revised ASD used in the V1.5 processing is more appropriate for describing OMPS/LP measurements.

Figure 12 shows relative differences between the mean LP and SAGE profiles using the same latitude bins shown in Fig. 11. The absolute value of the relative differences between LP

V1.5 and SAGE III is < 10% for 19–29 km, demonstrating good agreement. The relative differences are larger below 18 km, likely due to uncertainties in LP aerosol retrievals at these altitudes.

Figure 13 shows a scatter plot of individual zonal mean extinction values from each data set between 20.5 km and 25.5 km, selected for collocation within 10° latitude between 45° S and

60° N for the entire comparison period. Linear regression fits between OMPS/LP and SAGE III extinction data show a clear improvement in correlation coefficient from SAGE III vs. V1.0 to



SAGE III vs. V1.5 ($r = 0.83$ to $r = 0.97$), with a concurrent reduction in standard deviation σ for the V1.5 fit by a factor of two. These results give further quantitative evidence that the gamma function ASD is appropriate for OMPS/LP aerosol extinction retrievals.

**6. Summary and Conclusions**

This paper describes the derivation of a revised aerosol size distribution function to retrieve aerosol extinction profiles from OMPS/LP limb scattering radiance measurements. We use results from the CARMA microphysical model as a basis for the revised ASD to take advantage of CARMA's large range of particle size information. We find that using an ASD based on a gamma function fit (designated V1.5) requires fewer free parameters than our previous choice of a bimodal lognormal ASD (designated V1.0), and is more consistent with the CARMA particle size results. Evaluation of LP observed radiances is complicated by the measurement geometry (typically backward scattering in the SH, forward scattering in the NH) and the corresponding variation in phase function, as well as variations in scene reflectivity. The V1.5 ASD improves the performance of radiance-based retrieval algorithm internal validation tests, including reducing the magnitude of residuals between calculated and measured radiance and spectral dependence.

We also evaluated our revised ASD by comparing V1.5 retrieved extinction profiles to SAGE III measurements during June-December 2017. Relative differences between collocated zonal mean profiles are less than 10% between 19-29 km, with increased differences below 18 km. Regression fits to all data between 20-25 km show a better correlation coefficient between SAGE III data and LP retrievals with the V1.5 ASD ($r = 0.97$), and a factor of two improvement in standard deviation compared to results using the previous V1.0 ASD. We anticipate using the extensive CARMA model results in the future to determine additional ASDs for LP retrievals that can better represent natural aerosol variability in latitude, altitude, and season. CARMA results can also be used to develop a more effective ASD for aerosol retrievals following a volcanic eruption.

**Appendix A. Fitting Aerosol Size Distributions to OPC data**

One of the longest and most comprehensive records of local stratospheric aerosol conditions comes from the University of Wyoming's optical particle counters (OPC) carried on weather

balloons at Laramie, Wyoming, USA (41° N) at altitudes up to 30 km. The instrument measures
the number of aerosol particles in several size bins, ranging from 0.15 μm to 2 μm radius. In
most cases, a bimodal lognormal size distribution (BD) is used to fit OPC data if there are
enough different particle sizes measured. For background stratospheric conditions, however,
5    OPC data may not provide sufficient information about smaller particles ($r < 0.15$ μm) to
determine a robust BD fit.

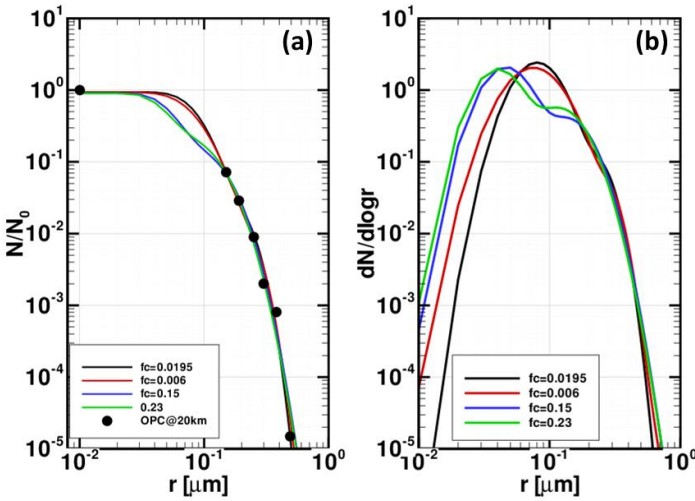

**Figure A1.** Estimated bimodal lognormal cumulative distributions (a) and differential
10   distributions (b) for nonvolcanic OPC measurement on 12 April 2000 at 20 km (Kovilakam and
Deshler, 2015). Measurements are shown as black dots on the left panel.

**Table A1.** Four BD fits to OPC data measured at Laramie Wyoming on April 12, 2010 at 20 km.

|  | ASD_1 | ASD_2 | ASD_3 | ASD_4 |
|---|---|---|---|---|
| Coarse mode fraction, $f_c$ | 0.0195 | 0.006 | 0.15 | 0.23 |
| Mode radius, $r_i$ (μm) | 0.080,0.238 | 0.075,0.280 | 0.046,0.140 | 0.040,0.120 |
| Mode Width, $\sigma_i$ | 1.45,1.25 | 1.56,1.21 | 1.45,1.43 | 1.43,1.47 |
| Angström exponent, AE | 2.45 | 2.40 | 2.40 | 2.40 |
| Effective radius, $r_{eff}$ (μm) | 0.1332 | 0.1335 | 0.1437 | 0.1470 |

15   An example of this situation is illustrated in Fig. A1, which shows four bimodal lognormal
distribution fits to the same OPC data at altitude 20 km, all having a similar Angström exponents





(AE) of approximately 2.4, but each with a different values of coarse mode fraction $f_c$. The fitted parameters as well as the calculated AE and $r_{eff}$ are given in Table A1. This topic is discussed further by Nyaku et al. (2018).

All the four fits are equally good to the OPC data but differ from each other significantly in the radius range between 0.01 μm to 0.1 μm because the gap in OPC size bins limits the ability to constrain a fit. As a consequence, the differences between the ASDs near 0.1 μm lead to significant changes in P(Θ) for backward scattering conditions, as shown in Fig. A2. It can be seen that P(Θ) is quite sensitive to the value of dN/dlog$_{10}$r around $r = 0.1$ μm when Θ > 90°. Larger values of P(Θ) in this range are closer to a Rayleigh scattering behavior.

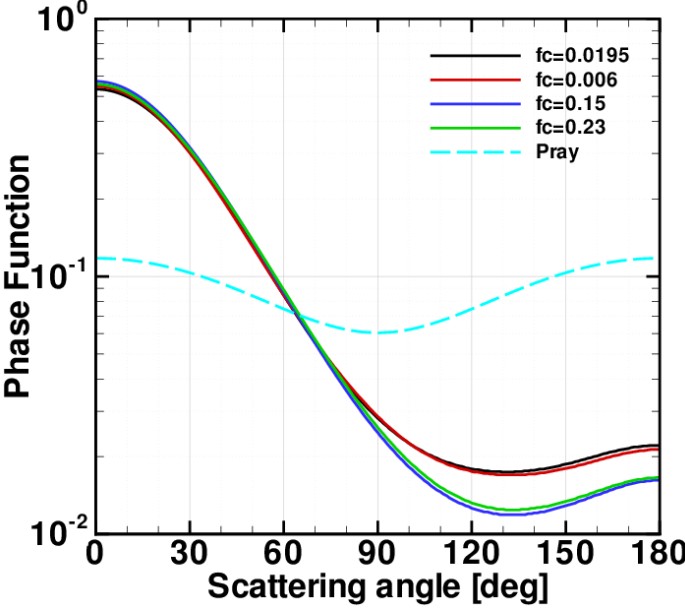

**Figure A2.** Aerosol phase functions at 675 nm as a function of single scattering angle for the four ASDs listed in Table A1.

**Acknowledgements**

We thank the OMPS/LP team at NASA Goddard and Science Systems and Applications, Inc. (SSAI) for help in producing the data used in this study. We also would like to thank Tong Zhu for her technical support. Z. C. and M. D. were supported by NASA contract NNG17HP01C.




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





**Table 1.** Bimodal size distribution used in OMPS/LP version 1 aerosol extinction retrieval

| AE | $r_{eff}$ (μm) | $f_c$ * | $r_i$ (μm) | $\sigma_i$ |
|---|---|---|---|---|
| 2.0 | 0.14 | 0.003 | 0.09, 0.32 | 1.4, 1.6 |

* $f_c$ is the coarse mode fraction, which is the ratio of the number of particles of the coarse mode to the
total number of particles for a bimodal lognormal distribution (Loughman et al., 2018).



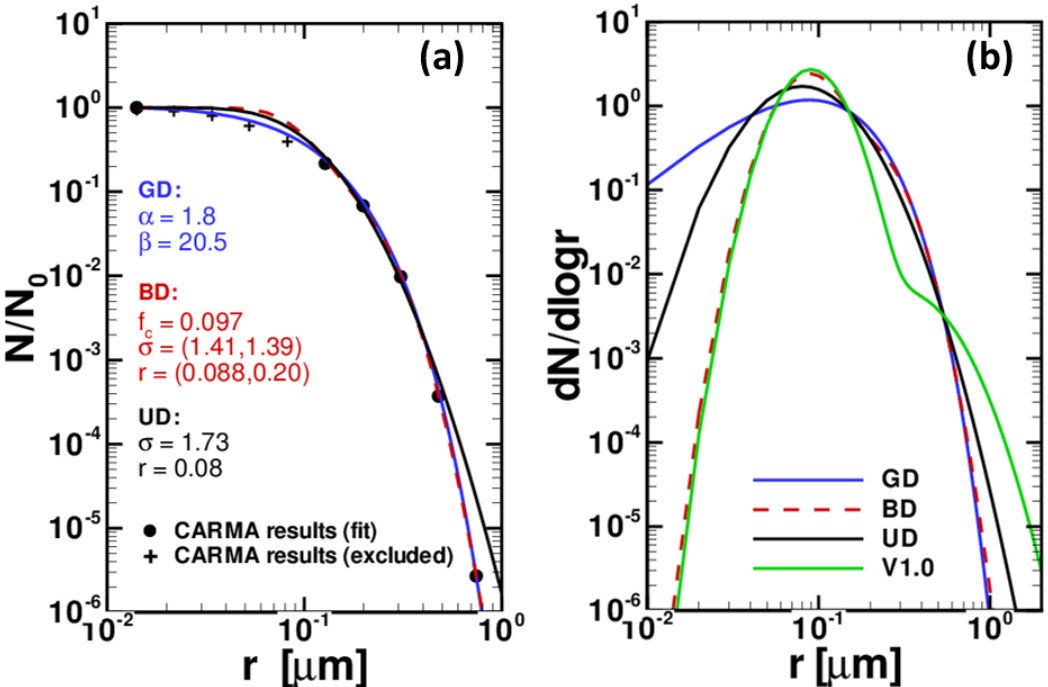

**Figure 1.** Comparison between Gamma size distribution (GD), bimodal lognormal size
distribution (BD) and unimodal normal distribution (UD). (a) Three cumulative size distribution
fits as a function of particle radius to the normalized CARMA data at 20 km. *Blue* = GD, *Red* =
BD, *black* = UD. The black dots represent cumulative CARMA data. Data points shown as (+)
were excluded from the fit. (b) Differential size distributions derived from the fitted parameters
shown in (a). For comparison, the size distribution used in V1.0 (green) is also shown. The V1.0
distribution has the largest *dN/dlogr* value at $r = 0.1$ μm and the smallest value at $r = 0.3$ μm.



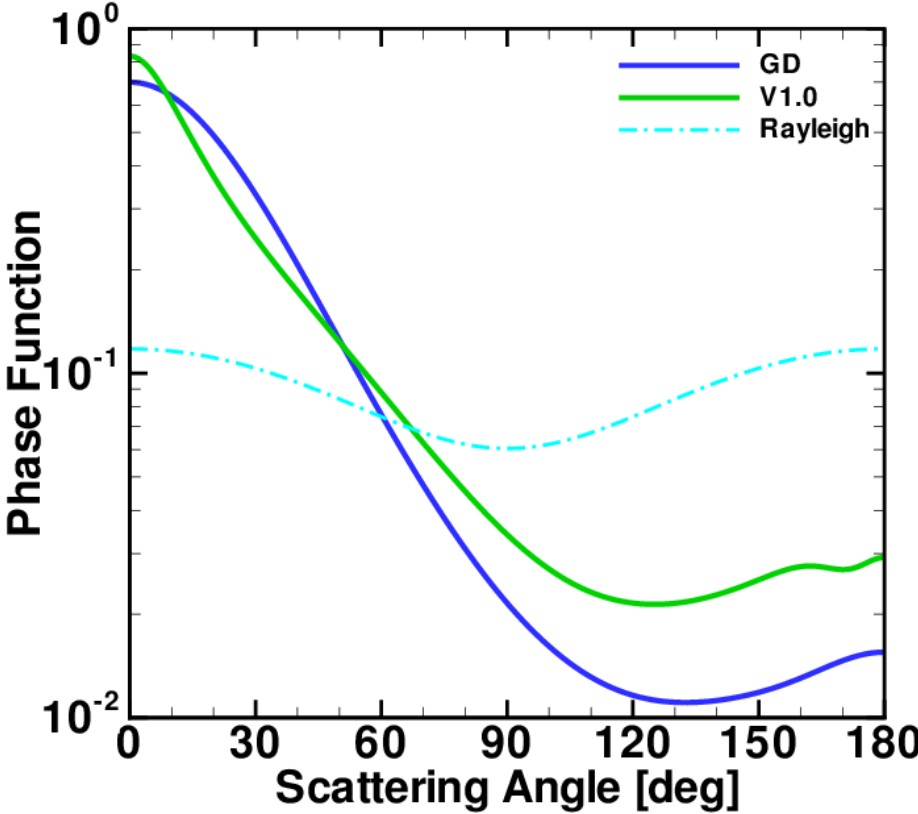

5      **Figure 2.** Comparison of 675 nm phase function P(Θ) between the GD (blue) and V1.0 (green)
size distributions. The Rayleigh phase function is also shown as a dashed line. The V1.0 P(Θ) is
closer to Rayleigh behavior at large scattering angles despite having more coarse particles ($r >$
0.5 µm) than others. This is because phase function at 675 nm is sensitive to particle radii at 0.1
µm as well as at 0.3 µm.



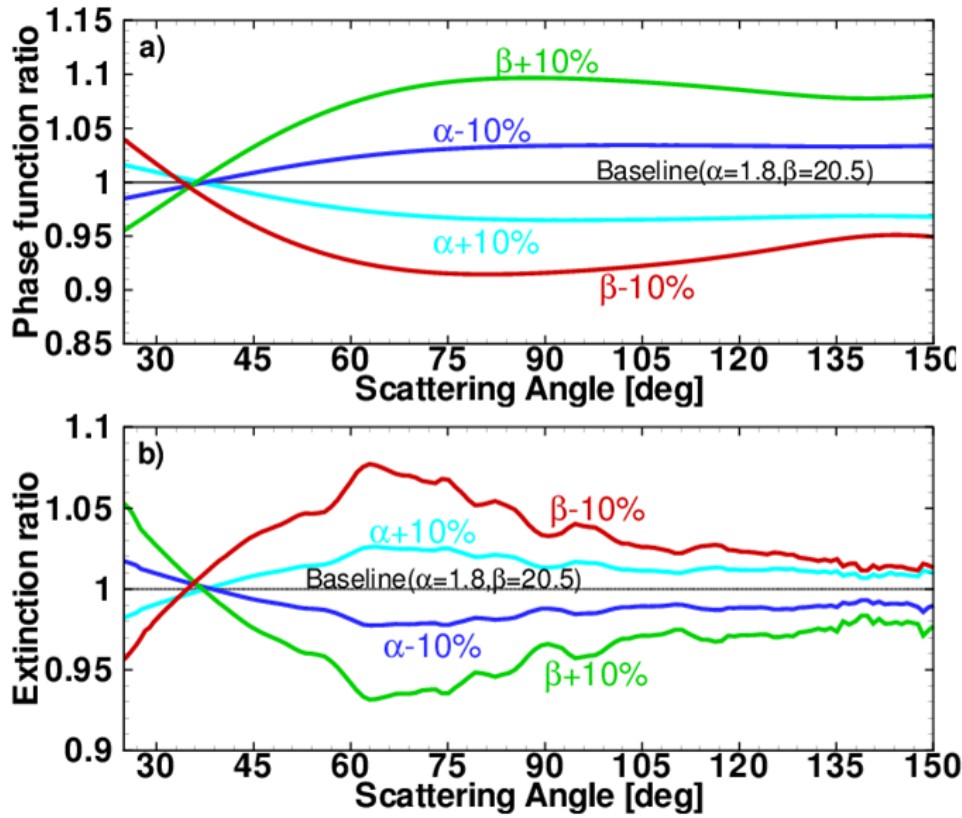

**Figure 3.** This figure shows how simulated phase function and retrieved extinction change when $\alpha$ and $\beta$ are perturbed by ±10% of the baseline values ($\alpha = 1.8$ and $\beta = 20.5$). All the phase functions and the extinctions shown are divided by the baseline data (black lines). (a) Ratio of the perturbed phase function to the baseline phase function. (b) Ratio of the perturbed extinction to the baseline extinction. The extinctions are retrieved using the simulated phase function and OMPS/LP measurements at 20.5 km for a single orbit on September 12, 2016. Note that the two curves are roughly anti-correlated, but the fractional change in extinction is about half of the change in P(Θ) depending on the single scattering angle Θ. (From Chen et al., 2018.)



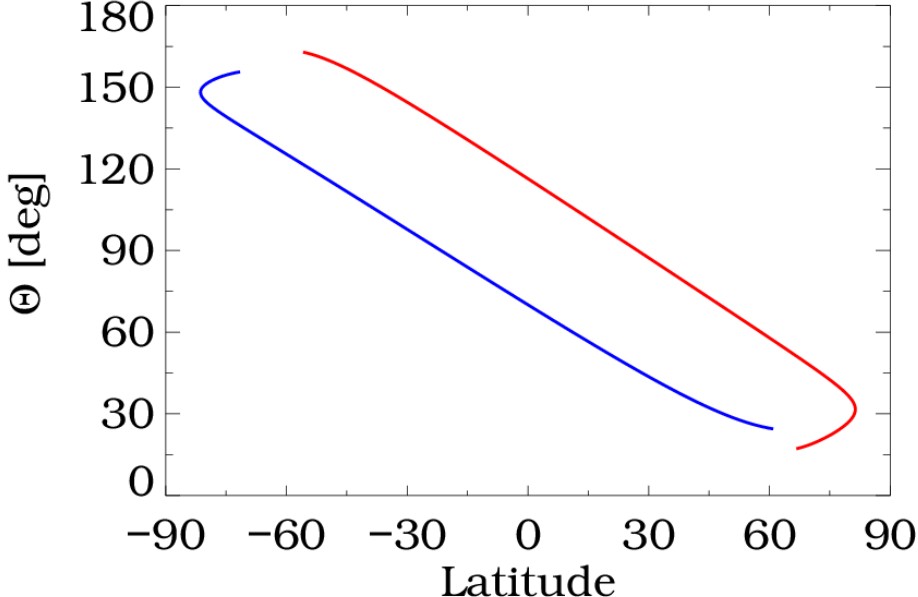

**Figure 4.** Variation of scattering angle (Θ) vs. latitude for OMPS/LP measurements on June 22
(red) and December 22 (blue). (From Loughman et al., 2018.)



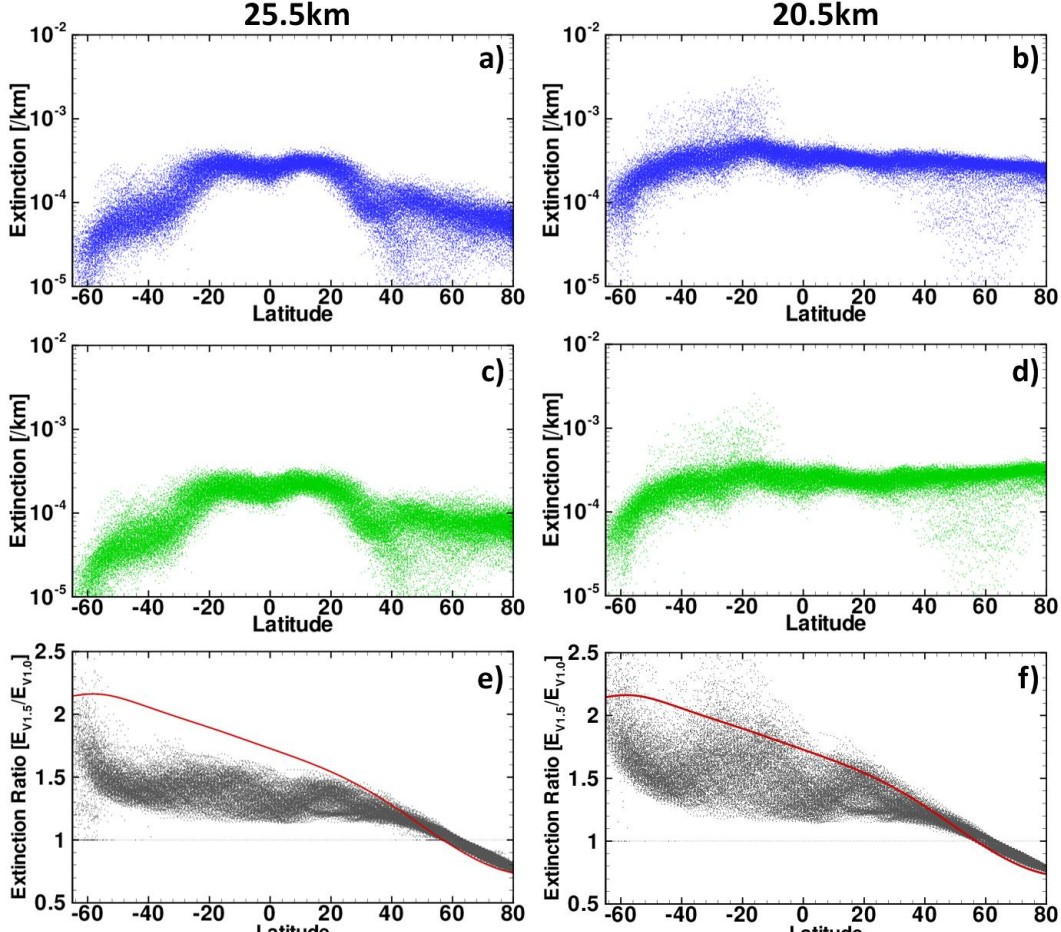

**Figure 5.** Scatter diagrams of retrieved aerosol extinctions for the V1.5 ASD (blue) and the V1.0 ASD (green) at 25.5 km (left panel) and 20.5 km (right panel) for entire month during the Calbuco period April 21 ~ May 20, 2015. The black dots in the bottom panel show extinction ratios ($E_{V1.5}/E_{V1.0}$), and the red lines shows the inverse of P(Θ) ratio ($P_{V1.0}Θ/P_{V1.5}Θ$). The ratio of extinctions has large variability at a given latitude, though the P(Θ) ratios do not. (From Chen et al., 2018.)



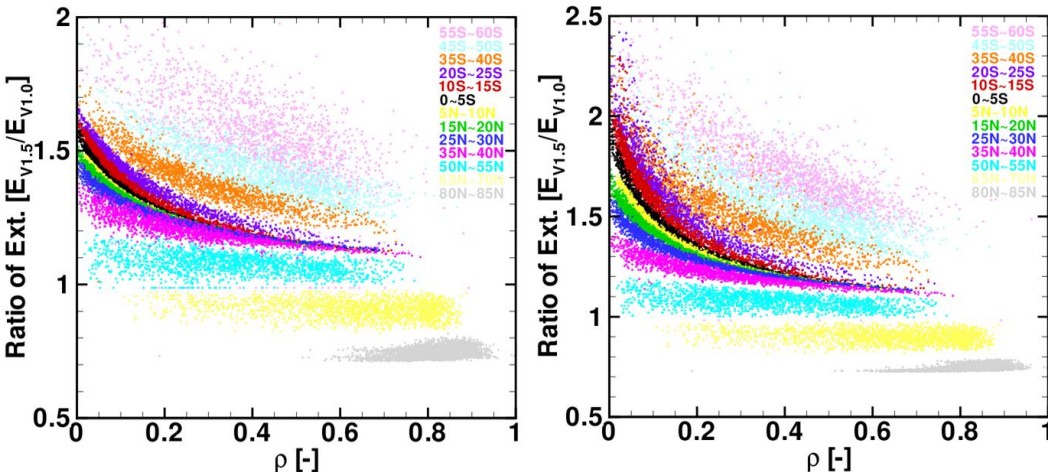

**Figure 6.** Scatter plots of extinction ratio ($E_{V1.5}/E_{V1.0}$) as a function of effective reflectivity ($\rho$)
for different latitude bins at 20.5 km (right) and 25.5 km (left). The figure shows that the
extinction ratios vary non-linearly with effective reflectivity, especially for reflectivity < 0.2.
The shape of the function changes considerably with latitude and altitude. (From Chen et al.,
2018.)



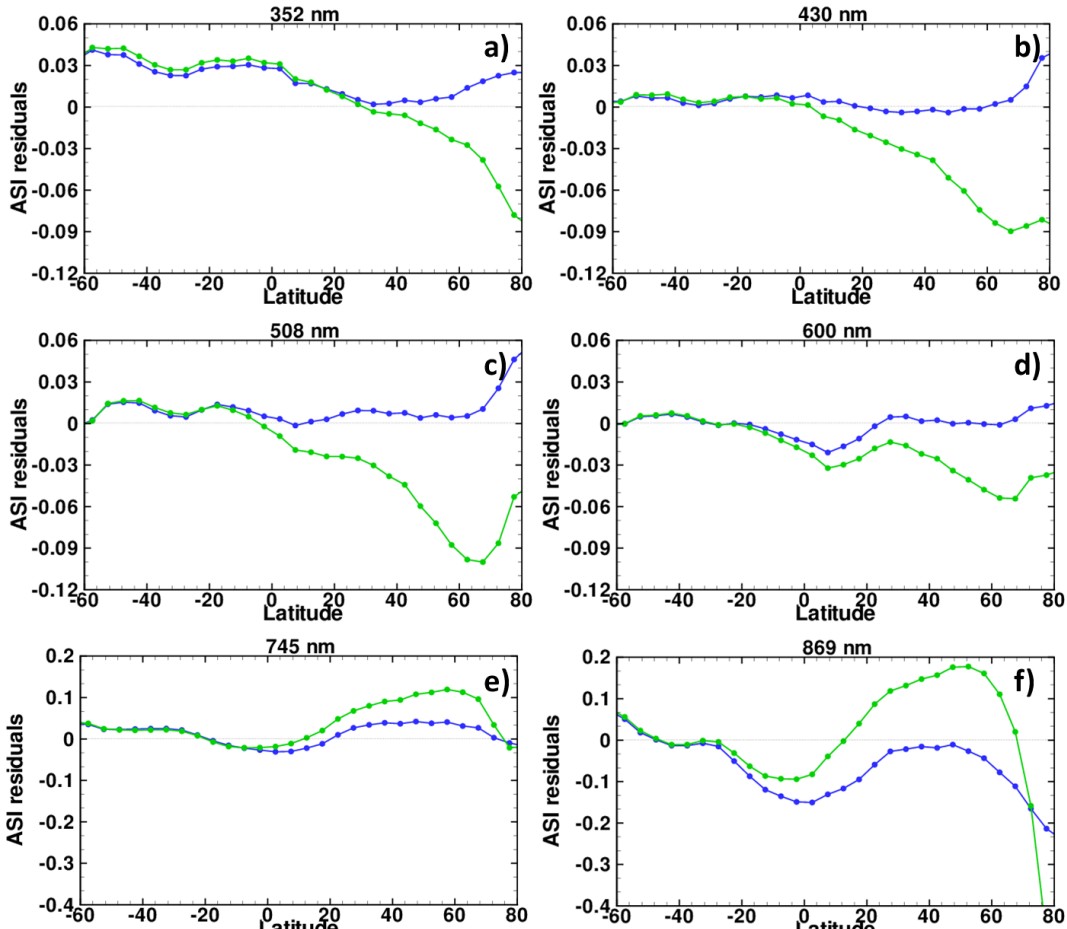

**Figure 7.** ASI residuals (ASIm - ASIc) at 20.5 km as a function of latitude for wavelengths at 352 nm, 430 nm, 508 nm, 600 nm, 745 nm and 869 nm. The V1.5 ASD (blue) does a better job in explaining the measured ASI relationship than the V1.0 ASD (green).





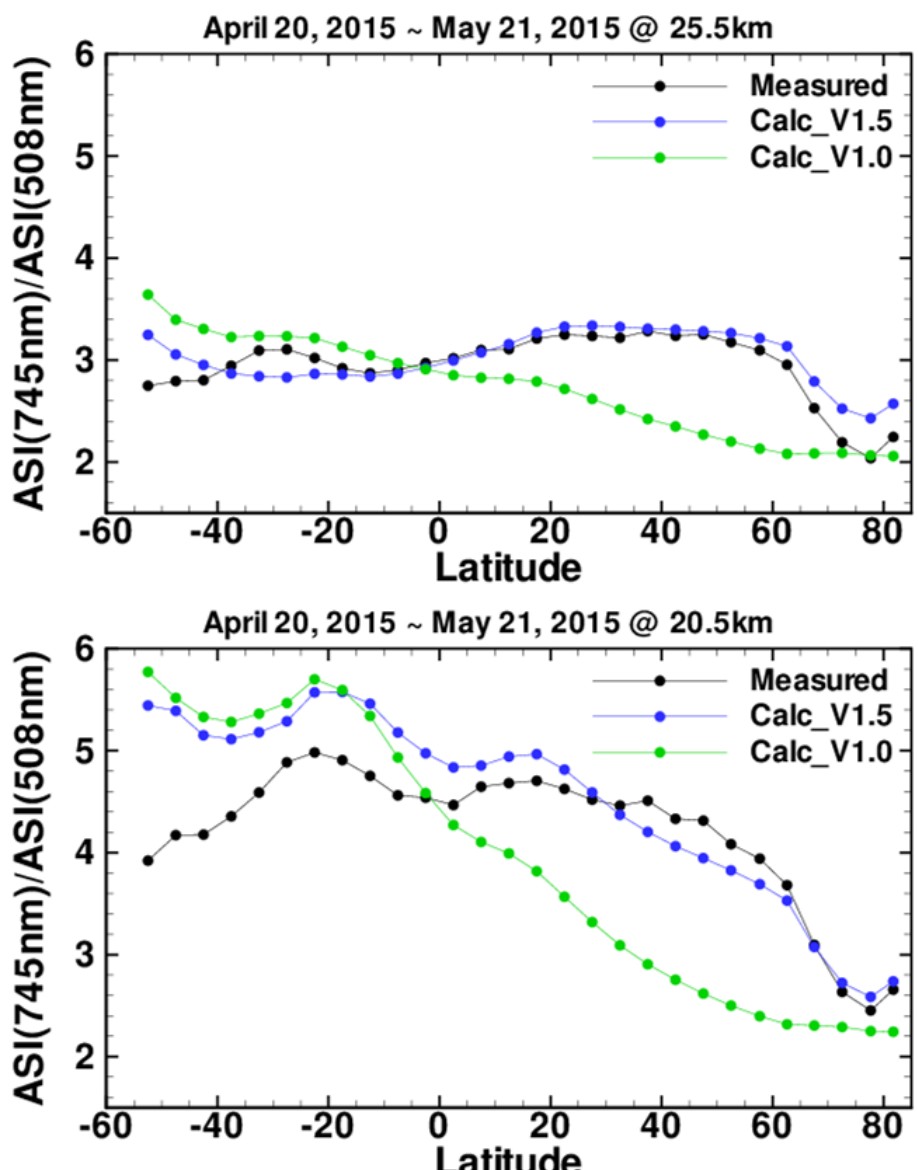

**Figure 8.** Zonal mean of ratio ASI(745 nm)/ASI(508 nm) at 25.5 km (top) and 20.5 km (bottom) as a function of latitude. Calculated values using the V1.5 ASD (blue dots) are more effective in explaining the measured ASI ratios (black dots) than the V1.0 ASD (green dots).




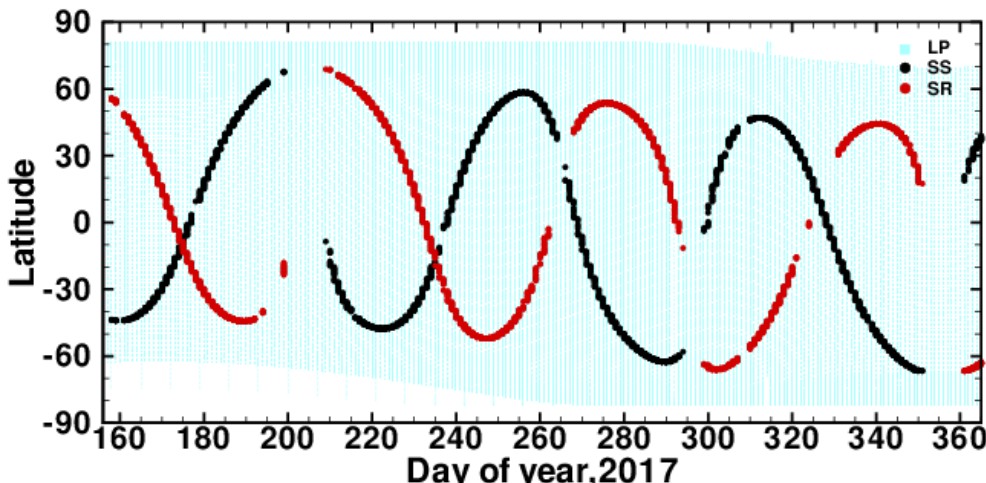

**Figure 9.** SAGE III/ISS solar occultation coverage compared to OMPS/LP coverage. Red: sunrises; black: sunsets; light blue: OMPS/LP.



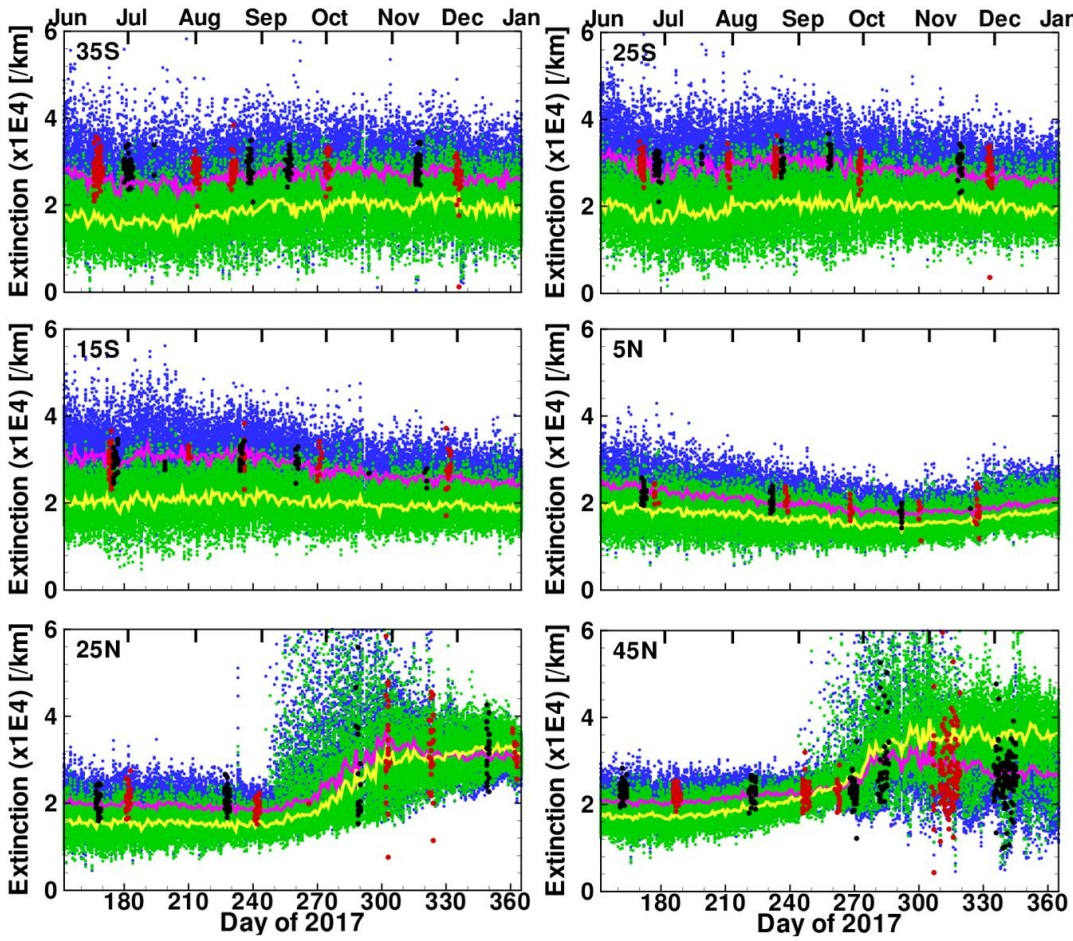

**Figure 10.** Time series of individual extinctions at 20.5 km observed by LP V1.5 (blue), V1.0 (green), SAGE sunrises (red) and SAGE sunsets (black) for six different ±5° latitude bins centered at 45° S, 25° S, 5° S, 5° N, 25° N, and 45° N during the comparison period. The pink and yellow lines show the median of aerosol extinctions at 20.5 km from LP V1.5 and V1.0, respectively.



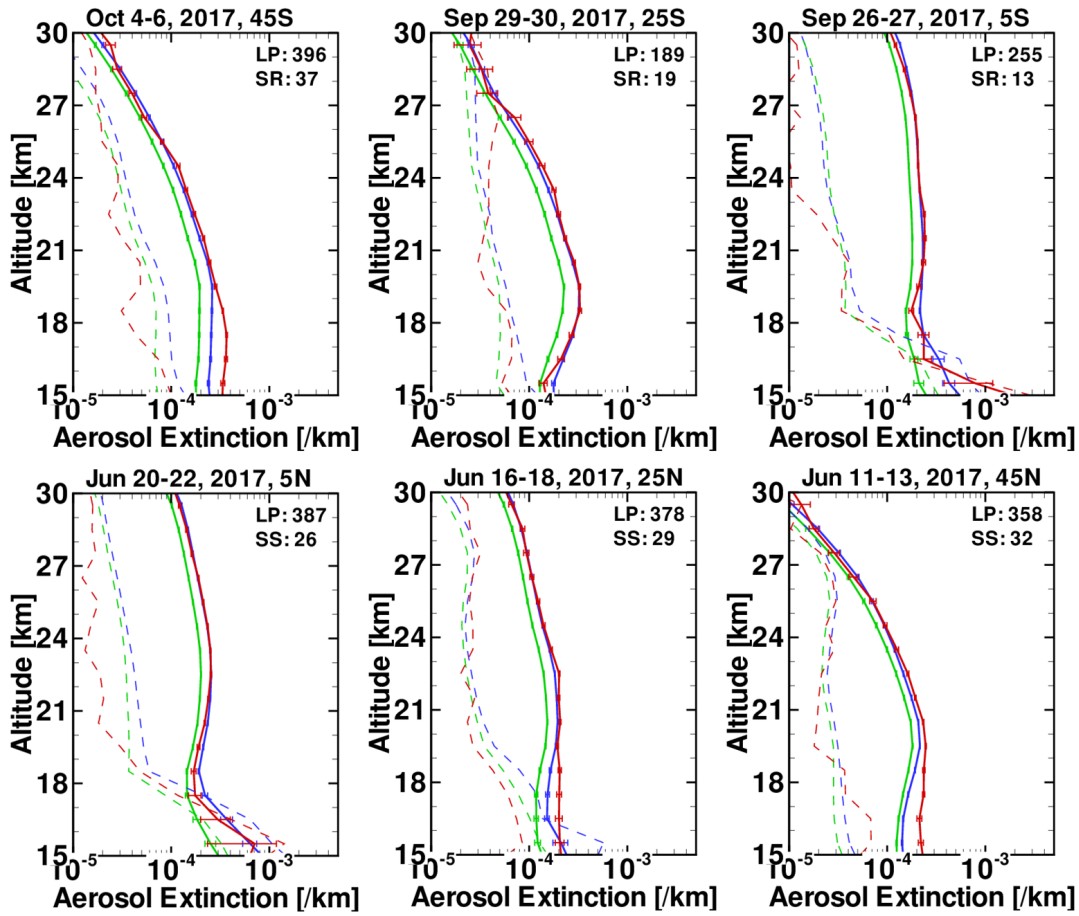

**Figure 11.** Comparison of zonal mean profiles of co-located SAGE III/ISS (red solid lines), LP
Version 1 (green solid lines) and LP version 1.5 (blue solid lines) aerosol extinction profiles for
six different ±5° latitude bins centered at 45° S, 25° S, 5° S, 5° N, 25° N, and 45° N. The dashed
lines show the corresponding standard deviations. The horizontal error bars indicate standard
error of the mean, $\sigma/\sqrt{N}$. The numbers in the top shows the number of measurements averaged
for each profile.



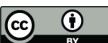

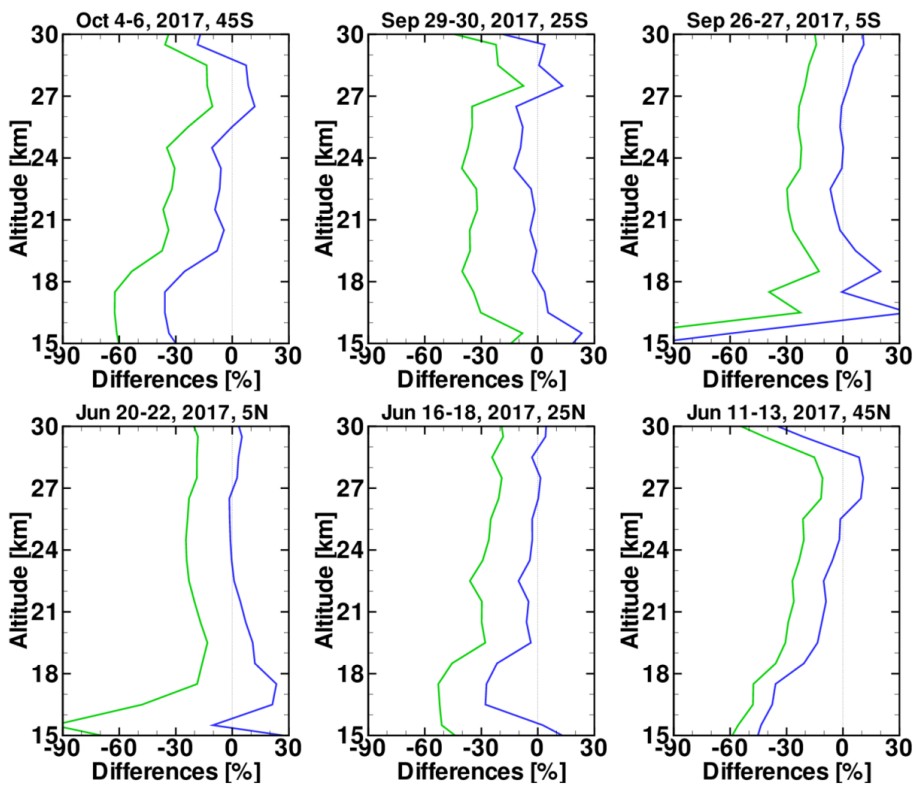

**Figure 12.** Relative differences of the mean aerosol extinction profiles between 675 nm OMPS/LP and 676 nm SAGE III/ISS. Difference=200×(LP-SAGE)/(LP+SAGE). Blue lines: LP V1.5 – SAGE; green lines: LP V1.0 - SAGE.





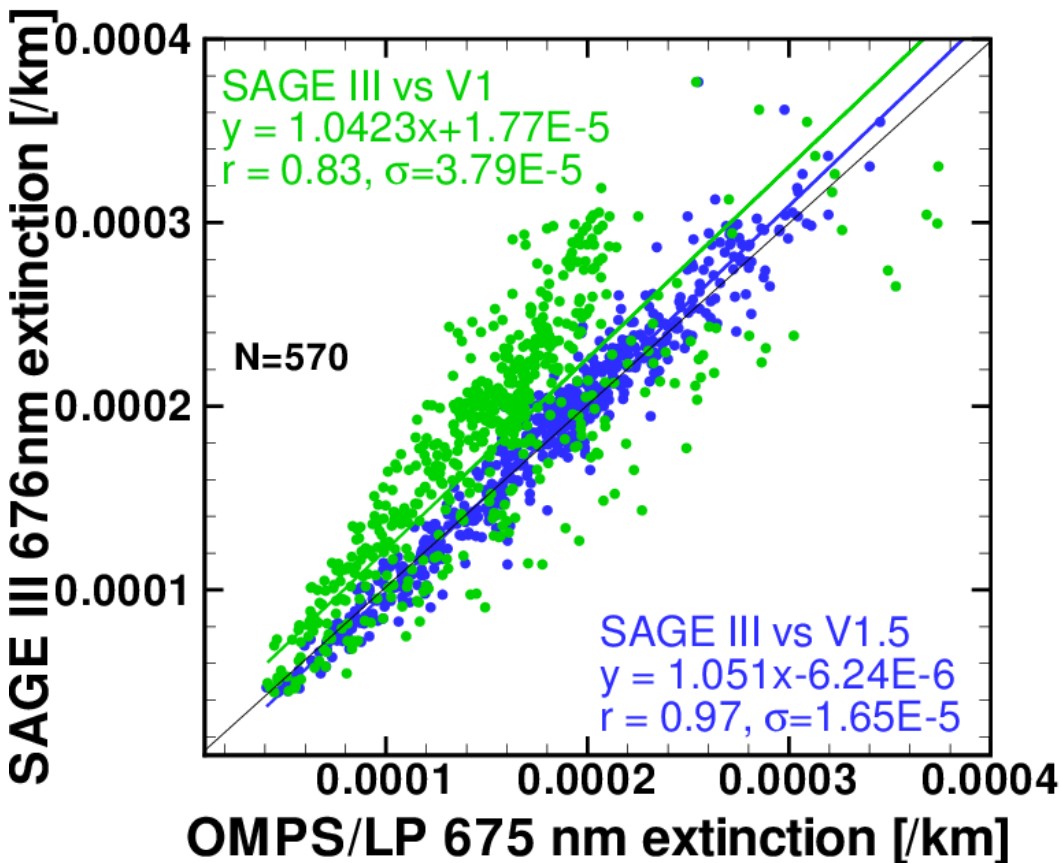

**Figure 13.** Correlation plot of SAGE III/ISS versus OMPS/LP V1.5 (blue) and SAGE III/ISS
versus OMPS/LP V1.0 (green) zonal mean aerosol extinctions in 10° latitude bins from 45° S to
60° N between 20-25 km for the entire comparison period. The blue and green lines show the
linear regressions between the data points, and the thin black line represents a 1:1 relationship.
The correlation coefficient r, the standard deviation σ, and the number of elements N used to
compute r are also shown.