# Peer review of "Improvement of stratospheric aerosol extinction retrieval from OMPS/LP using a new aerosol model"

_Atmospheric Measurement Techniques, 2018_

## Referee Comment (RC1) · Anonymous Referee #1 · 16 Aug 2018

General comments:

This is an interested and well written study on the improvement of stratospheric aerosol extinction profile retrievals from OMPS/LP limb scatter observations. Retrievals of this kind require a priori knowledge on the aerosol particle size distribution (or prior particle size retrievals/estimations), which is one of the main weaknesses of this retrieval method. The study uses a priori information on the aerosol particle size modeled with the CARMA model in combination with a gamma-function parameterization of the size distribution. In my opinion this study is a relevant contribution to the field and should be published with some modifications and additions.

[Figure]

I start with a few general comments, followed by some more specific and mainly minor comments:

I'm not a modeler myself, but hear from colleagues that modeling the size distribution of stratospheric aerosols is a non-trivial task and may be affected by different (partly arbitrary) assumptions. From that perspective, one can be skeptical whether modeled size distributions are in general suitable as a basis for aerosol retrievals from scattered radiance measurements. Perhaps the model simulations were also tuned to reproduce some observational data sets? I don't think this aspect is a big issue for the current paper, though, because the paper convincingly demonstrates that the retrieval results improve with the modeled size distribution. But I suggest adding a brief discussion stating that modeled size distributions may not (or probably will not) always lead to robust retrieval results.

Another more general point: The previous version of the OMPS/LP stratospheric aerosol data product was V1.0, the new version is V1.5. Please describe briefly, what was changed for the intermediate versions (1.1, 1.2, ..) that probably exist, too.

Specific comments:

Page 2, line 17: "in (Jaross et al., 2014)" -> "in Jaross et al. (2014)" (wrong cite command used) Page 3, line 9: "described in (Rault and Loughman, 2013)" -> "described in Rault and Loughman (2013)"

Similar typo in lines 11, 16 and 28 on page 3.

Page 3, line 29: "when the optical path along the line of sight (LOS) becomes thick". Can the "optical path" become "thick"? I suggest replacing "thick" by "optically thick"

Page 4, lines 19 and 20: please use the correct LaTex cite command (same problem as above)

Page 8, line 10: wrong citation command used.

[Figure]

Caption Fig. 1, last sentence: "The V1.0 distribution has the largest dN/dlogr value at r =0.1 $\mu$m and the smallest value at r =0.3 $\mu$m."

I don't understand the second part of this sentence. The V1.0 distribution does not have its smallest value at 0.3 micron. You probably mean compared to the other distributions, right? But this is not clearly stated.

Caption Fig. 2: The GD phase function shown is the one corresponding to the GD parameters in Fig. 1, right? I suggest to mention this.

Caption Fig. 5: "left/right panel" -> "left/right column" ?

Page 9, line 25: please use correct cite command

Page 9, line 28: "This behavior as a function of reflectivity is further illustrated in Fig. 6."

I'm not sure, if the behavior described in the previous sentence is really illustrated in fig. 6.? How does the Fig. illustrate that the retrieval is less sensitive to aerosols at lower altitudes? Please explain. It would also be good to provide a brief, qualitative explanation for the albedo dependence of the extinction ratios.

Page 11, line 9: please use correct cite command

Fig. 10: I'm not sure how to do this better, but in the current Figure a lot of the blue dots appear to be hidden by the green dots, i.e. the V1.5 LP values appear somewhat biased. Perhaps you can test plotting V1.5 on top of V1.0 and check, whether the apparent message of the Figure can be improved?

Page 12, line 25: I suggest writing "is generally < 10% for 19–29 km", because there are a few points in this altitude range, where the differences are larger than 10%.

Fig. 13: The Figure and the caption include a standard deviation. It is not clear – at least to me – which standard deviation this is.

Page 14, Table A1: Is the use of "mode radius" intended, or it this rather the median radius as in the main text of the manuscript?

Caption Fig. A1: I suggest defining f_c also in the figure caption.

---

## Referee Comment (RC2) · Anonymous Referee #2 · 9 Sep 2018

In this paper, the authors use a gamma distribution fit to a sectional aerosol model run (coupled to a GCM) to derive aerosol scattering phase functions. These are used in the radiative transfer forward model for the retrieval of aerosol extinction coefficient profiles from the limb scattering measurements made by the OMPS Limb Profiler. The assumption of the aerosol size distribution in the radiative transfer model for aerosol extinction retrieval from scattered light measurements is a long standing problem, and it boils down a basic lack of information in the remote sensing measurement to make a bias free retrieval. Other groups working on similar limb measurements with SCIAMACHY and OSIRIS have tried other forms of the size distribution with no real agreement or even criteria for what is best.

[Figure]

In general this is an insightful paper and the sample data sets that are presented show improvement over the previous version of the OMPS retrieval that used a bi-modal lognormal distribution. However, there are two related major points of concern. The first is that the main source of proof for improvement presented in the paper is analysis of a one month test data set (plus the six month time period used for SAGE III intercomparison). It does indeed seem that the gamma distribution is an improvement over the previous bi-modal assumption from V1.0; however, as the authors point out, the actual aerosol size distribution is a strong function of time, latitude, volcanic perturbation, etc., and it could very well be the case that the retrieval is worse at other time periods that are not analyzed. The V1.0 bi-modal assumption certainly had difficulties (choice of 5 free parameters, uncertainty in fitting OPC data) and the gamma function is demonstrably better, however, what about the simple unimodal size distribution assumed by the OSIRIS and SCIAMACHY algorithms? These are also simple 2 parameter distributions that roughly match the (measured, not modelled) background aerosol state. The corresponding phase functions for these distributions should be compared to the gamma distribution used here, and a clear case made for the use of gamma distribution. Since the bias is such a strong function of solar scattering angle, which for OMPS is essentially a latitude dependence, it might be the case that a "better" choice for OMPS is not a better choice for an instrument in a different orbit. Overall, users of limb scatter aerosol products would benefit from uniformity in the algorithm choices between the various groups, or at least publications that show how/why the assumptions are different.

The second major point is a more philosophical point about the use of model data in the retrieval. The authors are not yet using space and time dependent model size distribution, but they allude to this work as the first step towards that plan. To do this the authors must make a convincing case that the information folded into the retrieval from the model size distribution makes the result substantially better in a way that is quantifiable. The bias resulting from uncertainty in the aerosol size distribution is a second order effect that can be understood and characterized in a relatively simple way. But

now will introducing a complicated spatially/temporally varying model distribution make enough improvement to push this uncertainty to a third order effect, or will it just modify the results so that the second order effect is harder to understand and characterize due to the complex nature of the input assumption? Again, I realize this is not the case for this paper, but anticipation of this as an obvious next step is worrisome. Some of these issues should at least be discussed and approached with caution, especially as with this paper they have chosen to move away from using in-situ measurements to using model output.

Finally as far as I can tell, this paper is essentially a revision of Chen et al., 2018, which was not submitted for final publication. It seems than that this paper should stand independently and not reference the previous discussion paper, although the editor should weigh in on this.

Other minor comments:

Page 2, 3rd paragraph: The SCIAMACHY and OSIRIS work must be better referenced and discussed to put this work in context. These groups have done much more work, especially with regard to the size distribution, since the papers that are briefly mentioned here.

Several places throughout the paper refer to results from internal validation tests. It seems that some of these are shown and some are not. Is this simply referring to testing of the algorithm performance without validation data from other instruments? If not, this language is frustrating and it leaves the reader wondering what is behind the scenes.

Why choose the CARMA simulation for no volcanic eruptions? If the goal is really to use representative model data, why not run as realistic a simulation as possible for the OMPS mission time frame (which is definitely influenced by small volcanic eruptions) and then choose the median or average distribution? Why do the authors then choose to analyze a one month period that is perturbed by the Calbuco eruption?

What does a 10% change in the gamma distribution parameters mean in terms of particle size? Is this realistic for a moderate volcanic eruption?

Is there any potential for stray light or calibration effects in the interpretation of the spectral residuals? Why does the southern hemisphere look extremely good, where most of the differences in the phase functions seem to be for backscatter angles?

The refractive index should be representative of hydrated sulfuric acid and referenced.

The symbol E for aerosol extinction is not standard. Why not k? Then E switches to x in Equation 3. "Extinction ratio" is usually used for the ratio of aerosol to molecular extinction. What is the point of the discussion of the non-linearity of the ratio of the data versions with reflectivity?

---

## Referee Comment (RC3) · Anonymous Referee #3 · 12 Sep 2018

General: Article compares the retrieval results using two different assumed aerosol distributions with OMPS/LP limb scatter radiance profiles. Comparisons are shown to be more consistent with the spectral dependence of OMPS/LP data as well as correlative aerosol extinction data from SAGE III/ISS. This information is worthy of publication. However, the contents of this manuscript seem similar to another submission by the same author to AMTD (doi.org/10.5194/amt-2018-4). For example, both papers compare the degree to which two aerosol size distributions (bi-modal vs. gamma) match OMPS/LP radiance spectra. And, three figures in amt-2018-221 are "from amt-2018-4". What is new here? So, this document should do a much better job of clearly conveying the information that is unique to this manuscript. Otherwise, I do not support

publication

Specific comments: 1. Page 3, line 20: need to italicize I0 2. Section 2, especially page 4 is confusing. Are you describing V1.0 or V1.5? If the only difference between V1.0 and V1.5 is the phase function, then you need to clearly state that. As is, page 4, line 5 reads as though the use of Chahine's method is unique to V1.5 and was not used for V1.0. Furthermore, line 16 describes the number of iterations used for V1.0. How does that compare to V1.5? 3. Page 5, line 27, "(with some probably large uncertainty)" is conjecture that serves no purpose. Suggest deleting it or supplying evidence to prove your claim. 4. Page 5, 29: suggest "...well, the..." 5. Page 6, why not use the CARMA bins directly? 6. Page 7, line 5: What values of rmin and rmax are used here? 7. Page 7, line 15: Was the solution for the BD case determined using the method given earlier in line 6? Appendix X, indicates there is sufficient freedom to fit a wide range of size distributions. 8. Page 7, line 16: Fig. 1 leads me to believe that point at 0.015 um is used in the fit, but the text states that values "between 0.01 um and 0.1 um" were not used. Need to correct the figure caption or the text in the body. 9. Page 7, line 24: I must be missing something, but I do not see how for the bimodal ASD has "the smallest dN/dlogr value at r= 0.3 um. On the figure the lowest dN/dlogr is at r = 0.015um and 1um. Similar comment for the caption in Fig. 2. 10. Page 11, line 13: The doi for the SAGE III/ISS V5 data is: 10.5067/ISS/SAGEIII/SOLAR_BINARY_L2-V5.0 for the binary, and 10.5067/ISS/SAGEIII/SOLAR_HDF4_L2-V5.0 for the HDF version. 11. Page 14, line 16: suggest "...all having similar Angstrom exponents..." 12. Figure 2: What data do you have to make the claim at the end of the paragraph?

---

## Author Comment (AC1) · 12 Oct 2018

**Reply to Reviewer 1**
**Z. Chen et al.**
zhong.chen@ssaihq.com

*General comments:*
*This is an interested and well written study on the improvement of stratospheric aerosol extinction profile retrievals from OMPS/LP limb scatter observations. Retrievals of this kind require a priori knowledge on the aerosol particle size distribution (or prior particle size retrievals/estimations), which is one of the main weaknesses of this retrieval method. The study uses a priori information on the aerosol particle size modeled with the CARMA model in combination with a gamma-function parameterization of the size distribution. In my opinion this study is a relevant contribution to the field and should be published with some modifications and additions.*

Authors would like to thank Reviewer 1 for reviewing the manuscript and providing constructive comments. Our responses to the specific comments are given below in regular font.

*I start with a few general comments, followed by some more specific and mainly minor comments: I'm not a modeler myself, but hear from colleagues that modeling the size distribution of stratospheric aerosols is a non-trivial task and may be affected by different (partly arbitrary) assumptions. From that perspective, one can be skeptical whether modeled size distributions are in general suitable as a basis for aerosol retrievals from scattered radiance measurements. Perhaps the model simulations were also tuned to reproduce some observational data sets? I don't think this aspect is a big issue for the current paper, though, because the paper convincingly demonstrates that the retrieval results improve with the modeled size distribution. But I suggest adding a brief discussion stating that modeled size distributions may not (or probably will not) always lead to robust retrieval results.*

Authors: While we find in this study a general improvement in the quality of the OMPS LP aerosol retrievals by adopting the physically based and self-consistent CARMA-produced particle size distribution, we acknowledge here that the use of this particular model is not intended as a definitive prescription for the OMPS LP algorithms. The model is of course subject to a variety of uncertainties in its own right, in terms of formulation and implementation of its physical algorithms, and generally speaking the modeling of the stratospheric aerosol particle size distribution and composition is non-trivial and a subject of ongoing work by a number of researchers. For example, the version of the model used here does not yet include the possible impacts of volcanic eruptions on the background particle distribution, and neither does it include the impacts of non-sulfate aerosols that may be important in the UTLS (e.g., organics). We recognize that to push the approach taken here further, for example, to use a model-based climatology of aerosol properties to define altitude- and location-specific properties to be used in the OMPS LP algorithms requires at a minimum a more complete implementation of the relevant physics in the model (i.e., addition of missing species) and a thorough and independent evaluation of its capabilities and quality.

We have also added the following text in Section 3 as a cautionary note.

"Determining the particle size distribution based on model results is a challenging task. The assumptions inherent in any complex model can sometimes require arbitrary choices that influence the calculated results. So the size distribution adopted here for LP V1.5 aerosol extinction retrievals may not yield equally good results in all situations."

*Another more general point: The previous version of the OMPS/LP stratospheric aerosol data product was V1.0, the new version is V1.5. Please describe briefly, what was changed for the intermediate versions (1.1, 1.2, ..) that probably exist, too.*
Authors: No intermediate data versions exist between V1.0 and V1.5. We have added the following text at the end of Section 2 to help clarify the distinction.

"The primary change introduced for the LP V1.5 aerosol retrieval algorithm is the revised particle size distribution described in Sect. 3. Other changes with less impact on the retrieved extinction values include the use of vector radiative transfer calculations and the implementation of intra-orbit tangent height adjustments as described by Moy et al. (2017). In addition, the V1.0 retrievals only allowed a factor of two change in extinction for each iteration and executed three iterations, rather than the larger values (factor of five change, four iterations) given in Loughman et al. (2018). Based on inspection of test results, we revised those parameters for the V1.5 algorithm to allow a factor of three change in extinction for each iteration and four iterations of the retrieval."

*Specific comments:*
*Page 2, line 17: "in (Jaross et al., 2014)" -> "in Jaross et al. (2014)" (wrong cite command used)*
Authors: Fixed. Thanks.

*Page 3, line 9: "described in (Rault and Loughman, 2013)" -> "described in Rault and Loughman (2013)" Similar typo in lines 11, 16 and 28 on page 3.*
Authors: Fixed.

*Page 3, line 29: "when the optical path along the line of sight (LOS) becomes thick". Can the "optical path" become "thick"? I suggest replacing "thick" by "optically thick"*
Authors: We have replaced "thick" by "optically thick". Thanks.

*Page 4, lines 19 and 20: please use the correct LaTex cite command (same problem as above)*
Authors: Fixed.

*Page 8, line 10: wrong citation command used.*
Authors: Fixed.

*Caption Fig. 1, last sentence: "The V1.0 distribution has the largest dN/dlogr value at r =0.1 _m and the smallest value at r =0.3 _m." I don't understand the second part of this sentence. The V1.0 distribution does not have its smallest value at 0.3 micron. You probably mean compared to the other distributions, right? But this is not clearly stated.*

Authors: We have revised the caption of Figure 1 to clarify this point. "Among these size distributions, the V1.0 function has the largest $dN/d\log r$ value at 0.1 μm, but the smallest $dN/d\log r$ value at 0.3 μm."

*Caption Fig. 2: The GD phase function shown is the one corresponding to the GD parameters in Fig. 1, right? I suggest to mention this.*
Authors: You are right. We have added a note to the caption in Fig. 1: "The phase functions derived from the V1.0 and GD are shown in Figure 2."

*Caption Fig. 5: "left/right panel" -> "left/right column" ?*
Authors: We have replaced "left/right panel" with "left/right column".

*Page 9, line 25: please use correct cite command*
Authors: Fixed.

*Page 9, line 28: "This behavior as a function of reflectivity is further illustrated in Fig. 6." I'm not sure, if the behavior described in the previous sentence is really illustrated in fig. 6.? How does the Fig. illustrate that the retrieval is less sensitive to aerosols at lower altitudes? Please explain. It would also be good to provide a brief, qualitative explanation for the albedo dependence of the extinction ratios.*
Authors: Thank you for pointing this out. We have revised the text to read as follows: "The relationship between the large variability in extinction ratio shown in Fig. 5 and variations in LER is further illustrated in Fig. 6. The dependence of the extinction ratio $(\beta_{a,V1.5}/\beta_{a,V1.0})$ on ρ can become non-linear at low reflectivity $(\rho < 0.2)$, and the slope of the linear portion of this figure $(\rho > 0.2)$ varies with latitude. The non-linear variation in extinction ratio at $\rho < 0.2$ clearly increases in magnitude when moving from 25.5 km to 20.5 km, showing the altitude dependence of the additional contribution from Rayleigh scattering."

*Page 11, line 9: please use correct cite command*
Authors: Fixed.

*Fig. 10: I'm not sure how to do this better, but in the current Figure a lot of the blue dots appear to be hidden by the green dots, i.e. the V1.5 LP values appear somewhat biased. Perhaps you can test plotting V1.5 on top of V1.0 and check, whether the apparent message of the Figure can be improved?*
Authors: We have revised the plotting sequence as suggested for clarity. We have also revised some of the text in Sect. 5 to bring out the key message for this figure more effectively.

*Page 12, line 25: I suggest writing "is generally < 10% for 19–29 km", because there are a few points in this altitude range, where the differences are larger than 10%.*
Authors: You are correct. We added "generally" in the sentence.

*Fig. 13: The Figure and the caption include a standard deviation. It is not clear – at least to me – which standard deviation this is.*

Authors: We clarified that in the text: "standard deviation of differences σ (defined as

$$\sqrt{\Sigma_{i=1}^{N}(\beta_{LP,i} - \beta_{SAGE,i})^{2}/(N-2)}\ )"$$

*Page 14, Table A1: Is the use of "mode radius" intended, or it this rather the median radius as in the main text of the manuscript?*
Authors: "Mode radius" has been changed to "Median radius".

*Caption Fig. A1: I suggest defining f_c also in the figure caption.*
Authors: Done. Thank you.

---

## Author Comment (AC2) · 12 Oct 2018

**Reply to Reviewer 2**
**Z. Chen et al.**
zhong.chen@ssaihq.com

*In this paper, the authors use a gamma distribution fit to a sectional aerosol model run (coupled to a GCM) to derive aerosol scattering phase functions. These are used in the radiative transfer forward model for the retrieval of aerosol extinction coefficient profiles from the limb scattering measurements made by the OMPS Limb Profiler. The assumption of the aerosol size distribution in the radiative transfer model for aerosol extinction retrieval from scattered light measurements is a long standing problem, and it boils down a basic lack of information in the remote sensing measurement to make a bias free retrieval. Other groups working on similar limb measurements with SCIAMACHY and OSIRIS have tried other forms of the size distribution with no real agreement or even criteria for what is best.*

We appreciate the referee's comments and provide point-to-point responses below in regular font.

*In general this is an insightful paper and the sample data sets that are presented show improvement over the previous version of the OMPS retrieval that used a bi-modal lognormal distribution. However, there are two related major points of concern. The first is that the main source of proof for improvement presented in the paper is analysis of a one month test data set (plus the six month time period used for SAGE III intercomparison). It does indeed seem that the gamma distribution is an improvement over the previous bi-modal assumption from V1.0; however, as the authors point out, the actual aerosol size distribution is a strong function of time, latitude, volcanic perturbation, etc., and it could very well be the case that the retrieval is worse at other time periods that are not analyzed.*

Authors: We now use a one-year test data set covering all of 2017 for the ASI residual analysis shown in Fig. 7 and Fig. 8, in addition to the 1-month data set following the Calbuco eruption used for Fig. 5 and Fig. 6.

*The V1.0 bi-modal assumption certainly had difficulties (choice of 5 free parameters, uncertainty in fitting OPC data) and the gamma function is demonstrably better, however, what about the simple unimodal size distribution assumed by the OSIRIS and SCIAMACHY algorithms? These are also simple 2 parameter distributions that roughly match the (measured, not modelled) background aerosol state. The corresponding phase functions for these distributions should be compared to the gamma distribution used here, and a clear case made for the use of gamma distribution.*

Authors: We now include the parameters for the OSIRIS and SCIAMACHY size distributions in Table 1, and show the corresponding phase functions in Figure 2. While we did not create test data sets using those size distributions, we note that their differences from the V1.0 phase function in Fig. 2 are in the same direction as the gamma distribution (i.e. lower value at backscattered angles), but smaller in magnitude. So we would expect that processing LP data with one of these unimodal size distributions would yield less change relative to our V1.0 product than the gamma distribution adopted for V1.5. The improved agreement with SAGE III data for V1.5 extinction data shown in Fig. 10-13 suggests that we would not want to adopt a size distribution that produces less change in extinction.

*Since the bias is such a strong function of solar scattering angle, which for OMPS is essentially a latitude dependence, it might be the case that a "better" choice for OMPS is not a better choice for an instrument in a different orbit. Overall, users of limb scatter aerosol products would benefit from uniformity in the algorithm choices between the various groups, or at least publications that show how/why the assumptions are different.*

Authors: We agree with the reviewer's comment regarding the instrument-specific nature of our choice of size distribution, and have added the following text at the end of Sect. 5.

"Because of the large variation of phase function with scattering angle (Fig. 2) and the strong dependence between scattering angle and latitude for OMPS LP (Fig. 4), the size distribution determined here is not necessarily the optimum choice for a satellite instrument with a different measurement geometry resulting from a different orbit."

*The second major point is a more philosophical point about the use of model data in the retrieval. The authors are not yet using space and time dependent model size distribution, but they allude to this work as the first step towards that plan. To do this the authors must make a convincing case that the information folded into the retrieval from the model size distribution makes the result substantially better in a way that is quantifiable.*

Authors: We agree with the reviewer's point. We feel that internal validation comparisons such as Fig. 7 and Fig. 8 provide a quantitative demonstration of the large-scale improvement in LP retrievals from the use of the gamma function size distribution. We anticipate using similar comparisons to validate refinements to this size distribution that incorporate variations as a function of latitude, altitude, or season.

*The bias resulting from uncertainty in the aerosol size distribution is a second order effect that can be understood and characterized in a relatively simple way. But now will introducing a complicated spatially/temporally varying model distribution make enough improvement to push this uncertainty to a third order effect, or will it just modify the results so that the second order effect is harder to understand and characterize due to the complex nature of the input assumption? Again, I realize this is not the case for this paper, but anticipation of this as an obvious next step is worrisome. Some of these issues should at least be discussed and approached with caution, especially as with this paper they have chosen to move away from using in-situ measurements to using model output.*

Authors: We understand and appreciate the reviewer's concern regarding the potential challenge of implementing a variable size distribution for LP aerosol retrievals. We anticipate moving only incrementally towards this concept, evaluating only one parameter (e.g. latitude dependence) in each step, and carefully examining possible impacts such as discontinuities in retrieved extinction profiles at the boundary between different size distribution functions.

*Finally as far as I can tell, this paper is essentially a revision of Chen et al., 2018, which was not submitted for final publication. It seems than that this paper should stand independently and not reference the previous discussion paper, although the editor should weigh in on this.*

Authors: A previous version of this paper was submitted to this journal as manuscript amt-2018-4. We prepared responses to the original reviews and returned the revised manuscript to the journal. The associate editor recommended further revision, and stated "… you might think about re-writing the manuscript (possibly including the comparison with SAGE data and the new

OPC data) and submitting anew to AMTD". The current manuscript (amt-2018-221) has been substantially revised in accordance with this editorial guidance. Specifically, we expanded on amt-2018-4 to provide extensive comparisons with SAGE III/ISS data that were not available at the time that manuscript was submitted. The present journal editor for this manuscript also recommended that "you cite the old submission in the new submission". We have added a brief explanation of the relationship between this manuscript and amt-2018-4 at the end of Sect. 1 to clarify the situation for the reader.

*Other minor comments:*
*Page 2, 3rd paragraph: The SCIAMACHY and OSIRIS work must be better referenced and discussed to put this work in context. These groups have done much more work, especially with regard to the size distribution, since the papers that are briefly mentioned here.*
Authors: We have added more recent references to OSIRIS and SCIAMACHY results in Sect. 1. We have also added text to discuss those results in Sect. 3.

*Several places throughout the paper refer to results from internal validation tests. It seems that some of these are shown and some are not. Is this simply referring to testing of the algorithm performance without validation data from other instruments? If not, this language is frustrating and it leaves the reader wondering what is behind the scenes.*
Authors: We have revised the text to clarify that the phrase "internal evaluation" does indeed refer to tests that do not require external data sets.

*Why choose the CARMA simulation for no volcanic eruptions? If the goal is really to use representative model data, why not run as realistic a simulation as possible for the OMPS mission time frame (which is definitely influenced by small volcanic eruptions) and then choose the median or average distribution? Why do the authors then choose to analyze a one month period that is perturbed by the Calbuco eruption? What does a 10% change in the gamma distribution parameters mean in terms of particle size? Is this realistic for a moderate volcanic eruption?*
Authors: To understand the quality of the present aerosol size distribution and to estimate the uncertainty associated with the retrieved aerosol extinction, we first perform the aerosol retrieval code runs for conditions without a significant volcanic eruption. This provides a baseline situation. To evaluate the performance of the presented aerosol size distribution, aerosol extinction profiles were retrieved from OMPS/LP measurements before and after the Calbuco volcano eruption to see if the volcanic eruption can be captured by the new model. Creating an averaged size distribution from a multi-year model data set (covering the full OMPS LP mission) would give us less confidence that the retrieved extinction profiles are representative of a specific situation, and less understanding of how to interpret changes in retrieved extinction in terms of size distribution changes.
We added the following text to address the effect of gamma distribution parameter changes on particle size: "Examination of the corresponding differential distribution curves (not shown) indicates that increasing $\alpha$ produces an increase in the peak $dN/d\log r$ value, whereas increasing $\beta$ shifts this peak to larger values of $r$."

*Is there any potential for stray light or calibration effects in the interpretation of the spectral residuals?*

Authors: Potential errors due to stray light or absolute calibration bias are addressed in part through the use of altitude-normalized radiances in constructing the ASI measurement vector. Jaross et al. (2014) discuss possible remaining altitude-dependent errors from these sources.

*Why does the southern hemisphere look extremely good, where most of the differences in the phase functions seem to be for backscatter angles?*

Authors: Absolute ASI values are lower in the Southern Hemisphere because of the smaller phase function values at backscattered angles and the LP measurement geometry, as shown in Fig. 2 and Fig. 4. This difference can be a factor of 5-10 at 20.5 km (see Fig. 16 of Loughman et al. (2018)), which will generally produce smaller absolute residuals. Relative residual values (normalized by zonal mean ASI) are more similar between hemispheres.

*The refractive index should be representative of hydrated sulfuric acid and referenced.*

Authors: We added text: "for hydrated sulfuric acid (Palmer and Williams, 1975)" with the reference.

*The symbol E for aerosol extinction is not standard. Why not k? Then E switches to x in Equation 3. "Extinction ratio" is usually used for the ratio of aerosol to molecular extinction.*

Authors: The symbol "$E$" for aerosol extinction has been changed to "$\beta_a$", as suggested. You are right, "extinction ratio" is usually used for the ratio of aerosol to molecular extinction. For the sake of convenience in comparing, we use "extinction ratio" in this paper.

*What is the point of the discussion of the non-linearity of the ratio of the data versions with reflectivity?*

Authors: Figure 5 shows a complex behavior that the ratios of extinction are smaller than the ratio of phase function and the ratios of extinction vary with altitude even though the ratios of phase function do not. The point of the discussion is to explain that the effective reflectivity causes this complex behavior.

---

## Author Comment (AC3) · 12 Oct 2018

**Reply to Reviewer 3**
**Z. Chen et al.**
zhong.chen@ssaihq.com

*General: Article compares the retrieval results using two different assumed aerosol distributions with OMPS/LP limb scatter radiance profiles. Comparisons are shown to be more consistent with the spectral dependence of OMPS/LP data as well as correlative aerosol extinction data from SAGE III/ISS. This information is worthy of publication.*
We appreciate the constructive comments offered. Point-by-point responses are numbered in the same order as the comments given in the review.

*However, the contents of this manuscript seem similar to another submission by the same author to AMTD (doi.org/10.5194/amt-2018-4). For example, both papers compare the degree to which two aerosol size distributions (bi-modal vs. gamma) match OMPS/LP radiance spectra. And, three figures in amt-2018-221 are "from amt-2018- 4". What is new here? So, this document should do a much better job of clearly conveying the information that is unique to this manuscript. Otherwise, I do not support publication*
Authors:  A previous version of this paper was submitted to this journal as manuscript amt-2018-4.  We prepared responses to the original reviews and returned the revised manuscript to the journal.  The associate editor recommended further revision, and stated "… you might think about re-writing the manuscript (possibly including the comparison with SAGE data and the new OPC data) and submitting anew to AMTD".  The current manuscript (amt-2018-221) has been substantially revised in accordance with this editorial guidance.  Specifically, we expanded on amt-2018-4 to provide extensive comparisons with SAGE III/ISS data that were not available at the time that manuscript was submitted.  The present journal editor for this manuscript also recommended that "you cite the old submission in the new submission".  We have added a brief explanation of the relationship between this manuscript and amt-2018-4 at the end of Sect. 1 to clarify the situation for the reader.

*Specific comments:*
*1. Page 3, line 20: need to italicize I0*
Authors: Fixed.

*2. Section 2, especially page 4 is confusing. Are you describing V1.0 or V1.5? If the only difference between V1.0 and V1.5 is the phase function, then you need to clearly state that. As is, page 4, line 5 reads as though the use of Chahine's method is unique to V1.5 and was not used for V1.0. Furthermore, line 16 describes the number of iterations used for V1.0. How does that compare to V1.5?*
Authors:  We have added text in Sect. 2 to clarify that the primary difference between the V1.0 and V1.5 algorithms is the particle size distribution.  We also address other changes, including the number of iterations and the maximum change in extinction allowed for each iteration.

*3. Page 5, line 27, "(with some probably large uncertainty)" is conjecture that serves no purpose. Suggest deleting it or supplying evidence to prove your claim.*

Authors: You are right. We deleted "(with some probably large uncertainty)".

*4. Page 5, 29: suggest ". . .well, the. . ."*
Authors: Done. Thank you.

*5. Page 6, why not use the CARMA bins directly?*
Authors: The reviewer is correct that the aerosol optical properties can be calculated based on a RTM for each bin of the size distribution. However, the Mie calculation in the current LP aerosol code requires an analytic aerosol mode, rather than bin data. We use an analytical model of aerosol particle size distribution which deals with the ASD as a mean of size spectrum to accurately fit a cumulative distribution function (CDF) on the binned data using Deshler's method (Deshler et al., 1993, 2003).

*6. Page 7, line 5: What values of rmin and rmax are used here?*
Authors: We updated the text to provide the values: "$r_{min}$ = 0.01 μm to $r_{max}$ = 3 μm."

*7. Page 7, line 15: Was the solution for the BD case determined using the method given earlier in line 6? Appendix X, indicates there is sufficient freedom to fit a wide range of size distributions.*
Authors: Yes, it was the same.

*8. Page 7, line 16: Fig. 1 leads me to believe that point at 0.015 um is used in the fit, but the text states that values "between 0.01 um and 0.1 um" were not used. Need to correct the figure caption or the text in the body.*
Authors: You are correct. It was a typo in the text. It should be between 0.02 μm and 0.1 μm". We corrected the typo in our new revision.

*9. Page 7, line 24: I must be missing something, but I do not see how for the bimodal ASD has "the smallest dN/dlogr value at r= 0.3 um. On the figure the lowest dN/dlogr is at r = 0.015um and 1um. Similar comment for the caption in Fig. 2.*
Authors: We have revised the caption of Figure 1 to clarify this point.
"Among these size distributions, the V1.0 function has the largest *dN/d*log*r* value at 0.1 μm, but the smallest *dN/d*log*r* value at 0.3 μm.".

*10. Page 11, line 13: The doi for the SAGE III/ISS V5 data is: 10.5067/ISS/SAGEIII/SOLAR_BINARY_L2-V5.0 for the binary, and 10.5067/ISS/SAGEIII/SOLAR_HDF4_L2-V5.0 for the HDF version.*
Authors: Thank you for the information. We replaced "available through NASA's Atmospheric Science Data Center" with "the doi for the SAGE III/ISS V5 data is: 10.5067/ISS/SAGEIII/SOLAR_BINARY_L2-V5.0 for the binary, and 10.5067/ISS/SAGEIII/SOLAR_HDF4_L2-V5.0 for the HDF version."

*11. Page 14, line 16: suggest ". . .all having similar Angstrom exponents. . ."*
Authors: Fixed, thank you.

*12. Figure 2: What data do you have to make the claim at the end of the paragraph?*

We have modified text in Appendix A to clarify this statement:

"ASD_1 (black) and ASD_2 (red) have larger values of *dN/dlogr* around r = 0.1 μm shown in Fig. A1. Larger values of P(Θ) derive from the two ASDs in this range are therefore closer to a Rayleigh scattering behavior."

---

## Author Response (AR2)

Dear Editor,

We appreciate your comments. We have included as much as possible our responses to the reviewers to improve the manuscript as you suggested and provide point-to-point responses below.

Ref. #1:
The response to the first question on the model assumption has been included in the revised ms.

Ref. #2:
The concern of the reviewer on the variable size distribution for retrievals has been included in the revised ms.

The point of the no-volcanic eruption simulation of CARMA has been included in the revised ms.

The straylight concern has been also included in the revised ms.

We have switched extinction symbol E to symbol k. In order to avoid confusion, we have also switched iteration number symbol k in Eq. 3 to n.

Ref. #3:
The  responses to point 5 and 7 have been used in the revised ms.

Kind regards,

On behalf of the co-authors
Zhong Chen